# Cryo-EM structures of the translocational binary toxin complex CDTa-bound CDTb-pore from *Clostridioides difficile*

Akihiro Kawamoto [1,2,8], Tomohito Yamada [3,8], Toru Yoshida [3,4], Yusui Sato[5], Takayuki Kato [1] & Hideaki Tsuge [3,6,7] ✉

Some bacteria express a binary toxin translocation system, consisting of an enzymatic subunit and translocation pore, that delivers enzymes into host cells through endocytosis. The most clinically important bacterium with such a system is *Clostridioides difficile* (formerly *Clostridium*). The CDTa and CDTb proteins from its system represent important therapeutic targets. CDTb has been proposed to be a di-heptamer, but its physiological heptameric structure has not yet been reported. Here, we report the cryo-EM structure of CDTa bound to the CDTb-pore, which reveals that CDTa binding induces partial unfolding and tilting of the first CDTa α-helix. In the CDTb-pore, an NSS-loop exists in 'in' and 'out' conformations, suggesting its involvement in substrate translocation. Finally, 3D variability analysis revealed CDTa movements from a folded to an unfolded state. These dynamic structural information provide insights into drug design against hypervirulent *C. difficile* strains.

*Clostridioides difficile* infection is a major cause of nosocomial diarrhea and mortality. In the United States, nearly half a million people are diagnosed with, and approximately 12,800 deaths are directly attributed to, *C. difficile* infections (Antibiotic Resistance Threats in the United States, 2019-CDC). In addition to producing two large glucosyl transferase cytotoxins, TcdA and TcdB, certain strains produce the binary toxin CDT[1–3]; for example, the hypervirulent 027/BI/NAP1[4,5] and 078/BK/NAP[6–8] strains. Whether CDT promotes virulence is debatable; however, it depolymerizes actin, leading to the formation of microtubule base protrusions that increase pathogen adherence[9]. Furthermore, CDT induces host inflammation through toll-like receptor 2-dependent signaling, suppressing the protective host eosinophilic response[10]. Thus, synergy between multiple toxins may be a major aspect of hypervirulence.

Some bacteria have binary toxin translocation systems. These are comprised of a substrate A-component that translocates across the membrane via a B-component of the toxin; the latter has been shown to be involved in binding to host cells, thereby allowing protein unfolding and translocation of the enzymatic A-component across the membrane and into the cytosol[11–14]. Among such binary toxins are *Clostridium perfringens* iota Ia and Ib, *Clostridium spiroforme* CSTa and CSTb, *Clostridium botulinum* C2I and C2II, and *Clostridioides difficile* (previously known as *Clostridium difficile*) CDTa and CDTb[15]. Their B-components exhibit significant sequence homology (Supplementary Fig. 1). The B-components of the *C. perfringens*, *C. difficile*, and *C. spiroforme* systems bind to the host cell's lipolysis-stimulated lipoprotein receptor (LSR)[16,17], after which the A–B complex is internalized by endocytosis. Then, the A-component is translocated via the B-component pore across the endosomal membrane into the cytosol[18], assisted by host-cell chaperones, including Hsp90 and Hsp70[19,20]. In the cytosol, the A-component covalently transfers an ADP-ribose to actin, causing F-actin depolymerization, cell rounding, and eventually

[1]Institute for Protein Research, Osaka University, Suita, Osaka 565-0871, Japan. [2]Japan Science and Technology Agency, PRESTO, Kawaguchi, Japan. [3]Faculty of Life Sciences, Kyoto Sangyo University, Kamigamo-motoyama, Kita-ku, Kyoto, Japan. [4]Department of Chemical and Biological Sciences, Faculty of Science, Japan Women's University, 2-8-1 Mejirodai, Bunkyo-ku, Tokyo 112-8681, Japan. [5]Analytical Instruments R&D Division, HORIBA, Ltd., Kanda Awaji-cho 2-6, Chiyoda-ku, Tokyo 101-0063, Japan. [6]Institute for Protein Dynamics, Kyoto Sangyo University, Kamigamo-motoyama, Kita-ku, Kyoto, Japan. [7]Center for Molecular Research in Infectious Diseases, Kyoto Sangyo University, Kamigamo-motoyama, Kita-ku, Kyoto, Japan. [8]These authors contributed equally: Akihiro Kawamoto, Tomohito Yamada. ✉e-mail: tsuge@cc.kyoto-su.ac.jp

cell death[21,22]. Structural and functional studies on the A-component have shown that the enzymatic component consists of two domains: a C-terminal domain with ADP-ribosyltransferase activity, and an N-terminal domain that binds the B-component[23–26]. The *Bacillus anthracis* anthrax toxin belongs to another group of binary toxins that include an enzymatic component (edema [EF] and lethal [LF] factors) and a protein translocation channel (protective antigen [PA])[27]. In contrasting binary toxin systems between groups, there are significant differences in both activity and structure of their enzymatic A-components; however, it is generally thought that their B-components have functional similarities.

Several structures of the CDTb oligomer have been reported. Anderson et al. reported an asymmetric dimer of heptamers, or short form, consisting of a pore state stacked with a non-inserted pre-pore state at 3.7-Å resolution (PDB ID: 6O2N, C7 symmetry), and a symmetric dimer of heptamers, or long form, consisting of two partial β-barrel states at 3.9-Å resolution (PDB ID: 6O2M, C7 symmetry)[28]. Xu et al. also reported the same short form structure at 2.8-Å resolution (PDB ID: 6UWR) using cryo-EM and at 3.7-Å resolution (PDB ID: 6UWI)[29] using X-ray crystallography. In addition, they reported another symmetric dimer of heptamers consisting of two pre-pore states at 3.1-Å resolution (PDB ID: 6UWT). In all of these heptameric dimer structures, the receptor binding D4 domain (D4II) forms a heptameric static ring that interacts with another heptameric static ring through hydrophobic loop–loop interactions (Leu772 and Phe774). These di-heptameric states are insufficient for CDT pore formation in the membrane and CDTa translocation across the membrane. In contrast, a CDTb heptamer has been reported with CDTa (missing the N-terminal 16 residues) in the presence of the LSR (PDB ID: 6V1S), which prevents heptamer dimerization through the D4II domains[30]. These structures, called pre-insertion states, are different from the pre-pore and pore states, representing a transition from pre-pore to pore conformations. However, in this structure the LSR and D4II domains show no visible density because of their flexibility. Information on the binding between CDTa and CDTb is limited because the resolution is up to 3.8-Å. Notably, the physiological state of the heptameric CDTb-pore and the process by which CDTb interacts with and transports CDTa across the cell membrane remain unclear.

Here, we report two cryo-EM structures of the CDTa-bound CDTb-pore, long and short, at 2.64- and 2.56-Å resolution, respectively. In both structures, one CDTa molecule binds to the heptameric CDTb subunit through its N-terminal domain. We describe the interaction between CDTa and CDTb based on the highest-resolution structure to date of the CDTa-bound CDTb-pore (short). Binding induces the unfolding and tilting of the first N-terminal α-helix of CDTa, in addition to conformational changes to one of the constriction sites in the CDTb-pore, called the NSS-loop, which grips CDTa. Assessment of the dynamic features of the NSS-loop revealed that they exist in two conformations: 'in' and 'out,' with their interconversion apparently essential for translocation. Finally, using 3D variability analysis of the structure of CDTa-bound CDTb-pore (short), we revealed a series of CDTa motions within the complex, during transitions from folded to unfolded states.

## Results

### Preparation of the CDTb heptameric pore

We used a solubilization protocol similar to that used for preparing the iota toxin pore with lauryl maltose neopentyl glycol (LMNG) to mimic the buried liposome environment[31]. This protocol differed from the CDTb-pore preparation protocol[28]. For Ib-pore preparation, the Ib oligomer was purified using density-gradient ultracentrifugation after pro-peptide cleavage by chymotrypsin, which induced oligomerization in 10% ethanol at a low concentration of LMNG. In the case of CDTb, chymotrypsin cleavage of the pro-peptide induced oligomerization in the absence of ethanol, but it caused dimerization of the

heptamers, as previously reported[28,29]. We view the dimer of heptamers as an artefact in the absence of a membrane lipid, because it was formed by concealing the hydrophobic region of the pore from a hydrophilic solvent.

Thus, CDTb-pores were prepared by pro-peptide cleavage followed by oligomerization in LMNG-containing buffer at 37 °C. In these conditions, LMNG should cover the CDTb hydrophobic regions, allowing its solubilization. We observed di-heptamer and heptamer formation by density gradient centrifugation (Fig. 1a, b). Proteolysis in LMNG-containing buffer, followed by incubation at 37 °C, did not change the ratio of di-heptamer to heptamer (Fig. 1c).

We also checked for changes in di-heptamer and heptamer formation in the presence (five-fold more) and absence of LMNG. However, we did not observe any obvious differences because the heptamer peak was very small (data not shown). Next, we sought to understand why CDTb tends to form di-heptamers. The heptamers interact hydrophobically via Leu772, Phe774, and Pro776[28] (Fig. 1d). We therefore assessed di-heptamer formation by wild-type (WT), F774L, and F774G of CDTb by density gradient centrifugation. For both F774L and F774G, the di-heptamer fraction decreased and the heptamer fraction increased (Fig. 1e, f). When WT is bound to LSR via F774[28], it should be noted that it competes with the formation of a non-functional di-heptamer.

To obtain CDTa-bound CDTb-pores, CDTa was added to purified WT CDTb-pore at a 3-fold molar excess. Unbound CDTa was not removed by a further purification step. The undiluted, CDTa-bound, CDTb-pore sample was applied onto a glow-discharged Quantifoil holey carbon grid (R1.2/1.3, Cu, 300 mesh), and vitrified using a Vitrobot Mark IV (Thermo Fisher Scientific). Because the number and the mono-dispersibility of particles seemed favored, data collection was performed by Titan Krios (Thermo Fisher Scientific). The collected 11,284 movies at 0.88 Å/pixel were subjected to single-particle analysis. Three-dimensional (3D) classification revealed three classes: one including a di-heptamer, and the other two including heptamer pores of CDTa-bound CDTb-pores (long and short) (Supplementary Fig. 2). The ratios of the di-heptamer, long pore, and short pore were approximately 29:24:47%, with particle numbers 102116, 83061, and 167398, respectively. Mapping the di-heptamer at 3.2 Å revealed that it consisted of a pore state stacked with a non-inserted pre-pore state, consistent with an earlier CDTb structure (PDB ID: 6UWR). However, CDTa molecules were observed in each heptamer at their tops and bottoms. Unfortunately, CDTa density was averaged out in the map even at C1 map calculation (Fig. 2 and Supplementary Fig. 2).

By contrast, CDTa-bound CDTb-pores (long and short) showed clear CDTa densities in their maps (Fig. 2). The final resolutions of the long and the short CDTa-bound CDTb-pores were 2.64 and 2.56 Å, based on the Fourier shell correlation (FSC) 0.143 cut-off[31,32]. Because to our knowledge, there have been no structures published of CDTa-bound CDTb-pores, we focused on two structures: long and short CDTa-bound CDTb-pores (Table 1). Detailed interactions between CDTa and CDTb were analysed using the CDTa-bound CDTb-pore (short), which was subjected to focused 3D classification around CDTa, with excellent map quality (Supplementary Fig. 3).

### Overall structures of long and short of CDTa-bound CDTb-pores

We solved two structures of CDTa-bound CDTb-pores (Fig. 3a–d): one with a long stem and the other with a short stem. The former contained one entire β-barrel stem; however, the final map calculation showed that the tip of this stem (residues 337–358) had very weak density; therefore, we modeled the structure with a partial β-barrel stem, excluding residues 337–358 (Supplementary Fig. 2d).

By contrast, the short structure showed no density at the tip of the β-barrel stem (residues 332–363) because of its flexibility. Thus, both the long and short structures have partial β-barrel stems, lacking β-barrel tip residues 337–358 and 332–363, respectively. The most

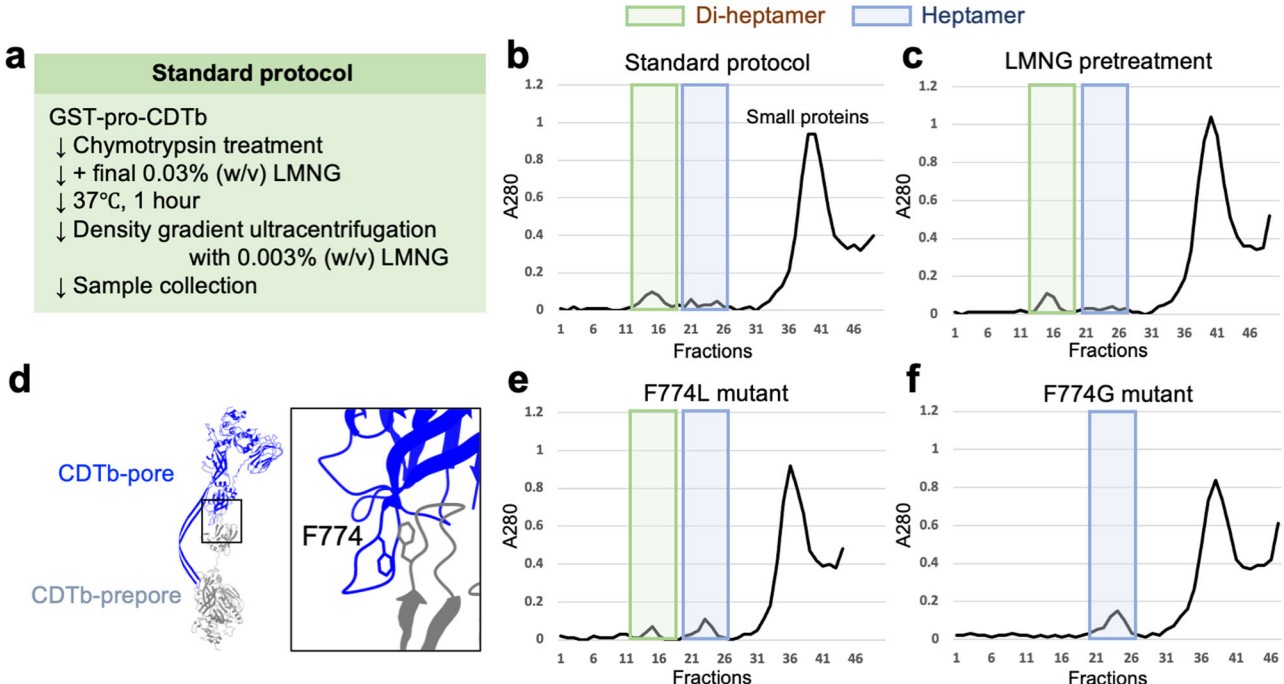

**Fig. 1 | CDTb purification. a** Protocol diagram. **b** Density gradient without LMNG pretreatment. **c** Density gradient with LMNG pretreatment before chymotrypsin treatment. **d** Location of CDTb residue F774 involved in di-heptamer formation. **e** Density gradient for the F774L mutant. **f** Density gradient for the F774G mutant. Both F774L and F774G were purified without LMNG pretreatment.

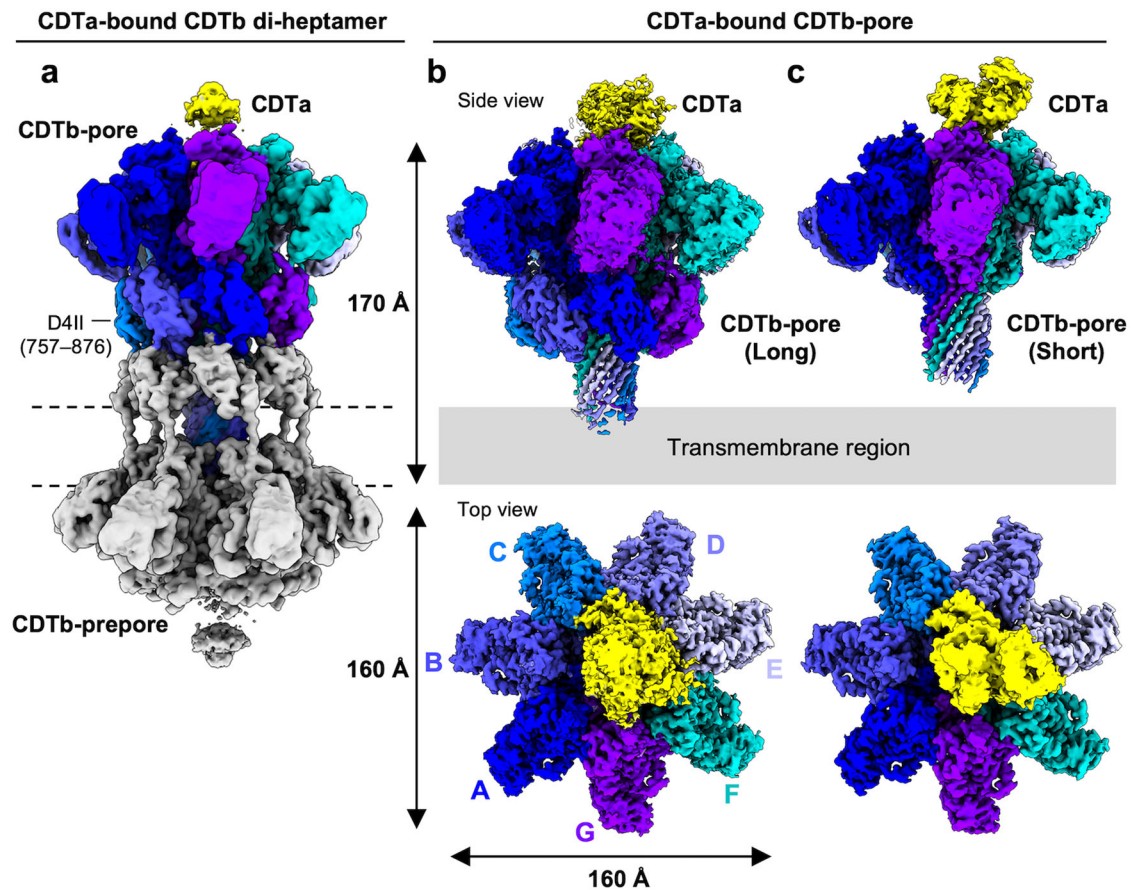

**Fig. 2 | Cryo-EM density maps of CDTa-bound CDTb-pores.** Transmembrane region is shown as dotted lines (left) or gray rectangle (right). **a** CDTa-bound CDTb di-heptamer. **b** Long CDTa-bound CDTb-pore. **c** Short CDTa-bound CDTb pore. Each protomer of the CDTb-pore and CDTa is colored.

**Table 1 | Cryo-EM data collection, refinement and validation statistics**

|  | Di-heptamer | Long | Short | Folded | Unfolded |
|---|---|---|---|---|---|
| EMDB ID | EMD-33188 | EMD-32043 | EMD-32041 | EMD-34136 | EMD-34137 |
| PDB ID | – | 7VNN | 7VNJ | 7YVQ | 7YVS |
| Data collection and processing |  |  |  |  |  |
| Magnification | 81,000 |  |  |  |  |
| Voltage (kV) | 300 |  |  |  |  |
| Electron exposure (e⁻/Å²) | 50 |  |  |  |  |
| Defocus range (μm) | –0.8 to –2.0 |  |  |  |  |
| Pixel size (Å) | 0.88 |  |  |  |  |
| Symmetry imposed | C1 |  |  |  |  |
| Initial particle images (no.) | 2,608,418 |  |  |  |  |
| Final particle images (no.) | 102,116 | 83,061 | 167,398 | 24,981 | 100,912 |
| Map resolution (Å) | 3.19 | 2.64 | 2.56 | 3.18 | 2.80 |
| FSC threshold | 0.143 | 0.143 | 0.143 | 0.143 | 0.143 |
| Map sharpening B factor (Å²) | –67.1 | –32.3 | –36.4 | –56.1 | –70.9 |
| Refinement |  |  |  |  |  |
| Model resolution (Å) | – | 3.2 | 2.8 | 3.4 | 3.0 |
| FSC threshold | – | 0.5 | 0.5 | 0.5 | 0.5 |
| Model composition |  |  |  |  |  |
| Non-hydrogen atoms | – | 38,823 | 30,661 | 30,742 | 30,593 |
| Protein residues | – | 4863 | 3856 | 3874 | 3856 |
| Ligands | – | 21 | 21 | 21 | 21 |
| B factors (Å²) |  |  |  |  |  |
| Protein | – | 61.64 | 93.72 | 90.02 | 78.04 |
| Ligand | – | 45.36 | 77.75 | 73.97 | 63.43 |
| R.m.s deviations |  |  |  |  |  |
| Bond lengths (Å) | – | 0.004 | 0.004 | 0.005 | 0.005 |
| Bond angles (°) | – | 0.949 | 0.956 | 0.977 | 0.990 |
| Validation |  |  |  |  |  |
| MolProbity score | – | 1.74 | 1.65 | 1.76 | 1.68 |
| Clashscore | – | 6.31 | 5.58 | 6.34 | 6.12 |
| Poor rotamers (%) | – | 0.37 | 0.40 | 0.75 | 0.98 |
| Ramachandran plot |  |  |  |  |  |
| Favored (%) | – | 94.27 | 95.01 | 93.78 | 95.09 |
| Allowed (%) | – | 5.73 | 4.99 | 6.22 | 4.91 |
| Disallowed (%) | – | 0.00 | 0.00 | 0.00 | 0.00 |

Table containing details of cryo-EM data collection, processing, and refinement including relevant statistics for all maps and models generated in this study.

striking difference between the structures was that D4II (receptor-binding domain II) was observed in the long, but not in the short complex. However, the two overall structures were similar, including a 1:1 binding ratio of CDTa:CDTb-pore.

CDTa contains two domains: an N-terminal CDTb-binding domain and a C-terminal ADP-ribosyltransferase domain (Fig. 3a). CDTb has four domains: 1' (W214–A296 or domain 1 without the pro-peptide), 2 (Y297–I513), 3 (S514–P616), and 4 (T617–D876) (Fig. 3b). The central pore body included domain 2, consisting parts designated 2c (residues 297–312 and 382–513) and 2 s (313–381). Domain 2 s is an extended β-hairpin, with seven copies assembled to form a membrane-spanning, 14-stranded, β-barrel. Domain 3 is between domains 1' and 2c, and domain 4 (D4I). Domain 4 is the receptor-binding domain at the outermost region of the pore, with two sub-domains, D4I (T617–L741) and D4II (T757–D876), joined through a linker (N742–P756), as seen in the long-form.

The lumen of the pore contained four constrictions (Fig. 3e, f). The first constriction includes a unique N-terminal di-calcium binding site, designated Ca-edge, with a 45-Å inner diameter. The second constriction includes the NSS-loop (residues 491–493), with a 23-Å inner diameter. The importance of both the Ca-edge and NSS-loop in CDTa-binding is described below. The narrowest constriction was formed by seven Phe residues (F455) from seven protomers with an inner diameter of 6-Å, designated φ-clamp and first described in the anthrax protective antigen (PA) pore[33]. The φ-clamp is a stable portion of the structure. The fourth constriction occurs at residue H314, located in the stem just below the φ-clamp. H314 is conserved between CDTb, Ib, and CSTb.

### CDTa-binding mode and translocational unfolding

In both the long and short complexes, CDTa was bound to the *cis*-side of the CDTb-pore through its N-terminal domain (Fig. 3c, d). Half of this domain was buried in the pore through several interactions with Ca-edges and NSS-loops. The interface area was estimated at 1960 Å². Five Ca-edges, from subunits C, D, E, F, and G, contributed to the interaction with the CDTa N-terminus (Fig. 3g).

Notably, four NSS-loops (C-F) were important for gripping CDTa. These interactions produced conformational changes in the loops. Our 2.56-Å resolution analysis provided more precise insights, revealing 'in' and 'out' loop conformations (blue and yellow, respectively, Fig. 4a). In the A, B, and G subunits, which had almost no interaction with the CDTa via the NSS-loop, the loop had two clear conformations. In the D and E subunits, the NSS-loop showed two conformations. However, in the C and F subunits, the NSS-loop was only observed in the 'out' conformation because of steric hindrance from CDTa.

We did not determine the pore structure in the absence of CDTa in this study; therefore, we checked the di-heptamer structure at 2.8-Å resolution (PDB ID: 6UWR), which is a complex of pore and pre-pore states. Interestingly, we found that both 'in' and 'out' states could be seen in the pore and pre-pore states in the cryo-EM maps, as described later. However, conformations deposited in PDB (ID: 6UWR) are 'in' for the pore and 'out' for the pre-pore states. Thus, we conclude that the NSS-loop conformation is in equilibrium between states in the default pre-pore and pore structures. Then, interactions with CDTa bias these loop conformations to fit or catch CDTa, especially in the C and F subunits.

The α-helices α1 and α2, at the N-terminus of CDTa, penetrate into the pores (red, Fig. 3e, g). Binding with CDTb induced partial unfolding of α1. The N-terminus included 19-WERKEAER-26 in the presented complex and 10-LKDKEKAKEWERKEAER-26 in the crystal structure (PDB ID: 2WN6); therefore, the nine N-terminal residues of α1 are unfolded (Fig. 5). This unfolding could be caused by steric hindrance between CDTa and CDTb. Moreover, α1 in the complex showed tilting at approximately 20° compared with the crystal structure of CDTa, suggesting that the tip of α1 was heading toward the φ-clamp (Fig. 5).

Next, we used 3D variability analysis (3DVA), an algorithm that fits a linear subspace model of conformational change to cryo-EM data at high resolution, to identify discrete conformational states of the short CDTa-bound CDTb-pore[34] (Fig. 6). Using a series of 20 frame maps, we constructed an animation of CDTa unfolding in the pore (Supplementary Movie 1). We obtained coordinates representing folded and unfolded CDTa in the complex (Fig. 6). Their features were: (1) The folded CDTa map shows a rigid cryo-EM density including the entire CDTa N-terminus (1-APIERPEDFL-). (2) The unfolded CDTa map shows not only tilting and unfolding of the first CDTa α1 in the complex but also the N-terminal density continuing to the φ-clamp. (3) Two NSS-loop conformations would adapt to fit transient translocational CDTa states: the unfolded and folded complexes showed different 'in' and 'out' conformations in the subunit D NSS-loop (Fig. 6h, i). The short CDTa-bound CDTb-pore map represents the total map, showing the small differences between folded and unfolded CDTa (Fig. 4a, b).

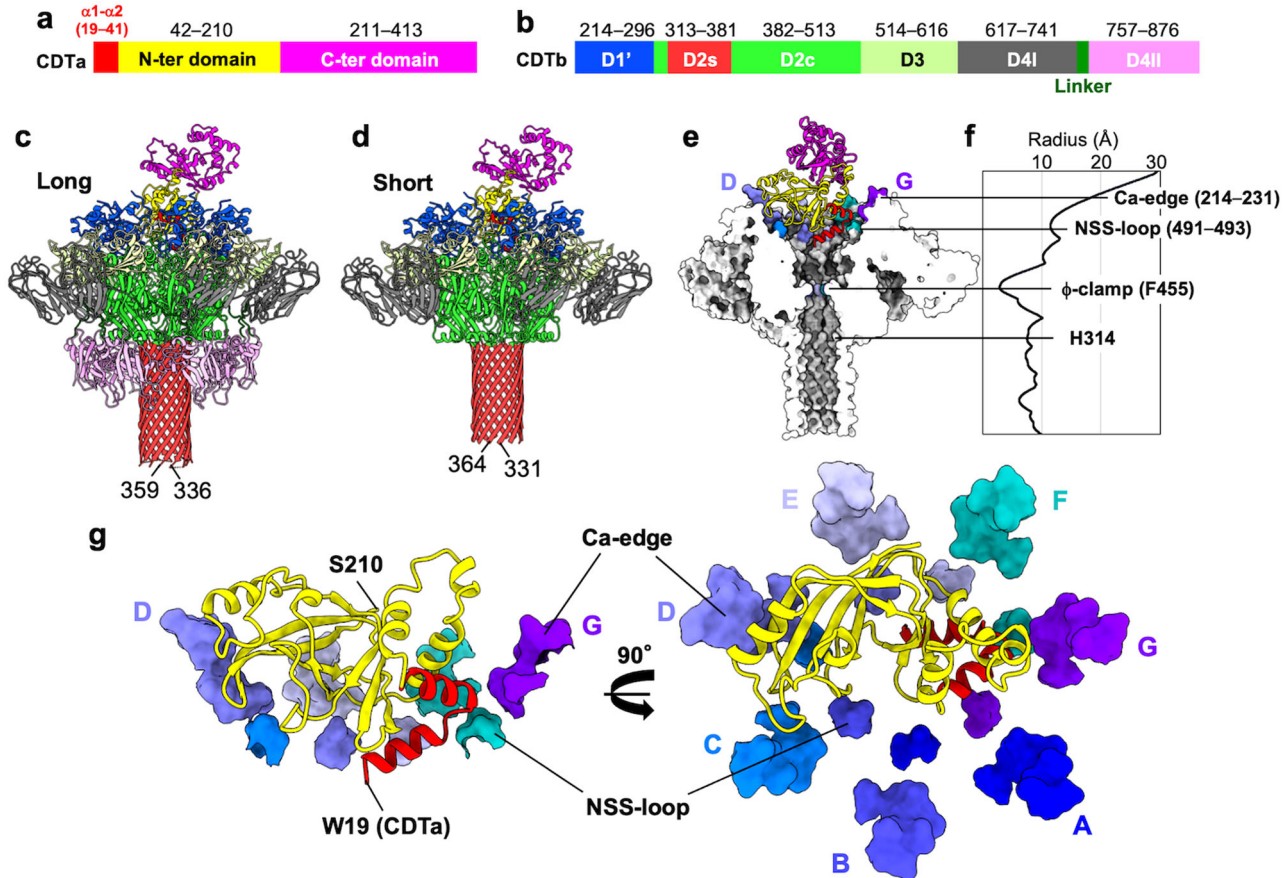

**Fig. 3 | Atomic structures of CDTa-bound CDTb-pores. a** Domain structure of CDTa. **b** Domain structure of CDTb after cleavage of the N-terminal pro-sequence. **c** Structure of long CDTa-bound CDTb-pore. **d** Structure of short CDTa-bound CDTb-pore. Colors are the same as those in panels **a**, **b**. **e** Cut-away view of CDTa-bound CDTb-pore. CDTb-pore is shown as a surface model, in which Ca-edges are colored by respective protomers. CDTa is shown as a ribbon. **f** CDTb-pore diameters measured from the CDTa-bound state (short) using HOLE software[53]. **g** Interaction between CDTa and CDTb. CDTa structure is shown as a ribbon. Constrictions (Ca-edges and NSS-loops) of the CDTb-pore are shown as surface models colored by respective protomers.

Considering the importance of the NSS-loop for CDTa binding and translocation, we sought to determine whether NSS-loop rigidity affects the CDTa binding using surface plasmon resonance (SPR). We made two mutants, NSS > SPS and NSS > PSS, in the loop, inserting a single proline residue to increase its rigidity[35]. Before and after endocytosis, CDTa binds to the CDTb-pore via Ca-edge at neutral and acidic pH, respectively. Thus, SPR data were obtained at pH 7.4 and 5.5 and at two $CaCl_2$ concentrations (Table 2, Supplementary Fig. 4): no calcium (chelated) and high (5 mM). At pH 7.4, the dissociation constant $K_D$ was smaller at high calcium due to the dissociation rate constant $k_d$. These data suggest that calcium stabilizes the CDTa-CDTb complex. Though WT and SPS yielded nearly identical results, the $K_D$ for PSS was slightly larger than those of WT and SPS. These data suggest that the rigidity of PSS rigidity affects CDTa-CDTb weak binding. On the other hand, at pH 5.5, though it was unexpected, the $K_D$ values for WT, SPS, and PSS were less than those at pH 7.4. For the PSS mutant, $K_D$ was greater than those of WT and SPS at the lower calcium concentration, though it was not apparent at 5 mM. Though PSS rigidity did not cause a major change in CDTa-CDTb binding, we speculate that its rigidity should affect translocation.

## Discussion

We obtained three different classes of CDTa-bound CDTb-pore from the sample prepared with LMNG. Though di-heptamers coexisted with single heptamers, the CDTa map was averaged out, even at C1 map calculation. Therefore, we determined two high-resolution structures of the CDTa-bound CDTb-pore. By analysing discrete conformational states of the short CDTa-bound CDTb-pore using 3DVA of cryoSPARC[34], we obtained one successive map from the short CDTa-bound CDTb-pore. The movie derived from that map (Supplementary Movie 1) shows how CDTa unfolds in the complex. We also refined two coordinates for folded and unfolded CDTa structures, represented in the first and the last states in the movie, respectively.

The long and short structures, with other reported structures, of CDTb, including di-heptamers, are summarized in Supplementary Fig. 5. Between the long and the short structures, the definitive difference is that D4II is clearly visible in the long form. That is, after stem formation, D4II is stably fixed around the stem. In di-heptamer structures, two different conformations of D4II were observed. These dynamic features of D4II are likely important in binding the host cell LSR. In our previous study, we confirmed that the receptor-binding domain is also involved in determining oligomerization efficiency, using a D4II-deficient mutant Ib[36].

*Cis* and *trans* interactions of the D4II receptor-binding domains have been reported to form di-heptamers of CDTb[28,29], with F774 being the key in the D4II-D4II *trans* interaction, as well as in the LSR interaction. We confirmed its importance by density gradient centrifugation (Fig. 1). This residue is a Leu in Ib; thus, in vitro, Ib forms predominantly heptamers not di-heptamers[37].

The overall structures of Ia-bound Ib-pores and CDTa-bound CDTb-pores are similar. Both share the Ca-edge and NSX-loop (Ib:NSQ-loop) constrictions for A-component binding.

Ca-edges are di-calcium binding sites that are DTDNDNIPDSYE in CDTb and DTDNDNIPDAYE in Ib. Di-calcium densities were clearly

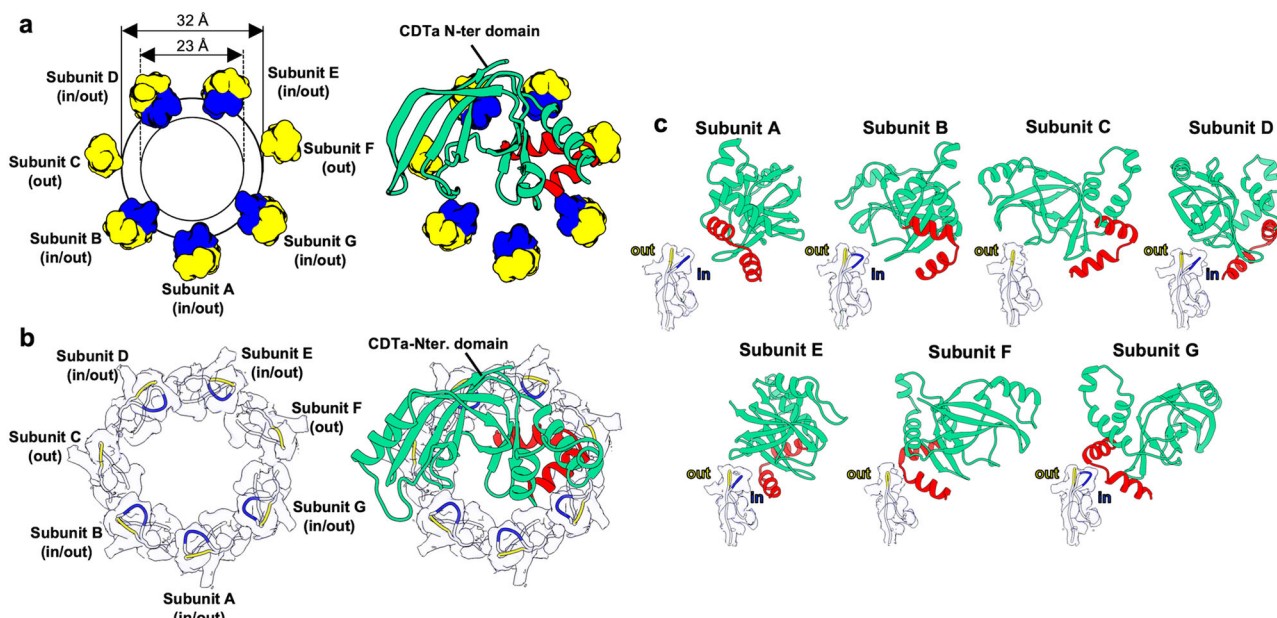

**Fig. 4 | NSS-loop conformations. a** Concentric circles of NSS-loops. Conformations in the CDTa-bound state are shown without (left) or with (right) the CDTa N-terminal domain. 'in' loop conformations are shown in blue, 'out' conformations in yellow. **b** Cryo-EM map of NSS-loops of the CDTb-pore in the CDTa-bound state without (left) or with (right) the CDTa N-terminal domain. **c** The close-up views of 'in' and 'out' conformations of subunits A–G.

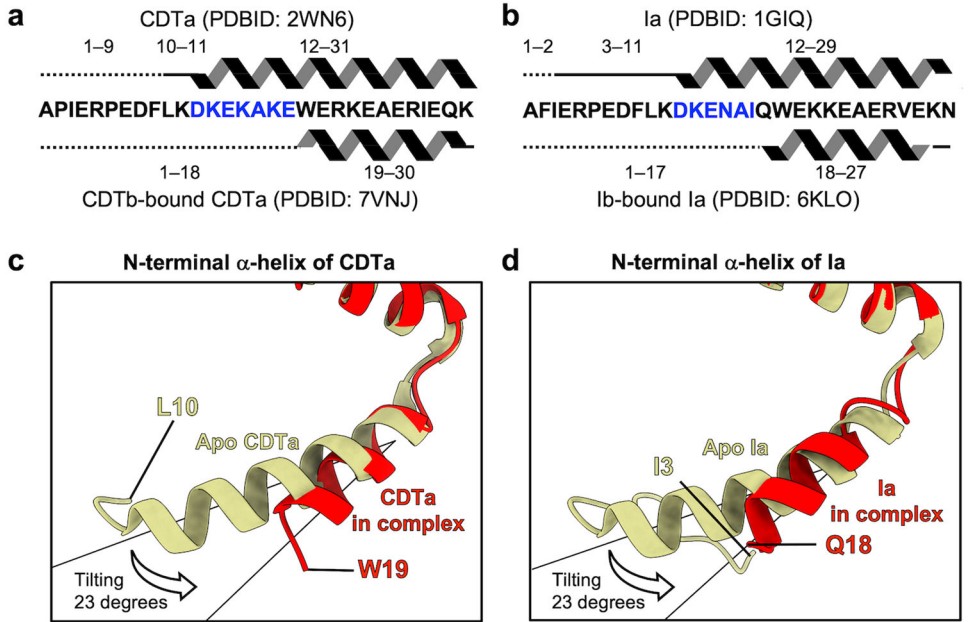

**Fig. 5 | N-terminal α-helix conformational changes in CDTa and Ia. a** Secondary structures of the CDTa crystal structure in the apo state (PDB ID: 2WN6) and the CDTa cryo-EM structure in the CDTb-pore complex (PDB ID: 7VNJ). **b** Secondary structures of the Ia crystal structure in the apo state (PDB ID: 1GIQ) and the Ia cryo-EM structure in the complex with Ib-pore (PDB ID: 6KLO). In both (**a**), (**b**), the dotted line indicates residues whose structures were not built. The thick line indicates residues without secondary structure. Residues of the α-helix unfolded in the complex are shown in blue. **c** Three-dimensional structures of the apo state (gold) and CDTb-bound CDTa (red). **d** Three-dimensional structures of the apo state (gold) and Ib-bound Ia (red).

present in both Ib and CDTb[37]. The φ-clamp is formed by seven Phe residues (F455 in CDTb and F454 in Ib) from seven protomers and has an inner diameter of 6-Å[37]. The conformations of Phe residues in CDTb and Ib were similar, in which these side-chain densities are visible. Although the NSX-loop is not perfectly conserved, it functions to grip the A-component in both CDTb and Ib, making it an essential structural element.

Although the NSS-loop is important for CDTa binding, the strong binding between CDTa and CDTb seems to inhibit the efficient

translocation of CDTa. To overcome these constraints, binding between CDTa and CDTb needs to be loosened and less specific during translocation. In this study, we found two conformations of NSS-loop, 'in' and 'out' (Fig. 4), indicating that two stable states–instead of many flexible states–exist in the NSS-loop. These two conformations were also observed in the map density in the pre-pore and pore structures of CDTb (Supplementary Fig. 6a, b). Although we assigned 'in' conformations to NSQ-loops of the Ib pore structure[37], we re-examined the Ib-pore and Ia-bound Ib-pore maps and structures (PDB IDs: 6KLX and

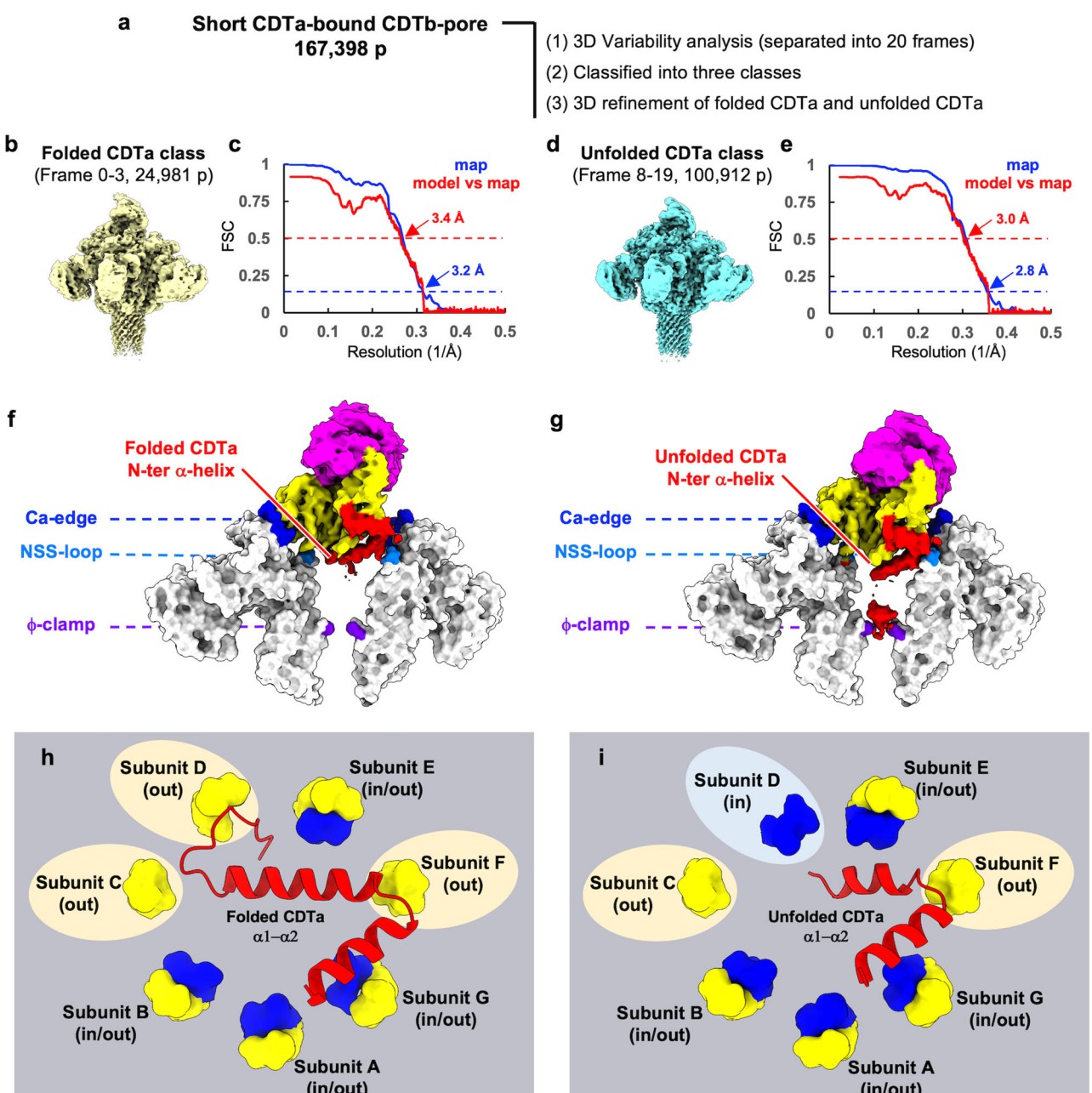

**Fig. 6 | 3D variability analysis from the short CDTa-bound CDTb-pore. a** Flow chart of cryo-EM image processing with 3D variability analysis. The short CDTa-bound CDTb-pore particles were classified into the folded CDTa class and the unfolded CDTa class, followed by 3D refinement respectively. More detailed information is described in result and method. **b** The density map of the folded CDTa class. **c** FSC curve of the half maps and FSC curve for cross-validation between the map and model of folded CDTa class. **d** The density map of the unfolded CDTa class. **e** FSC curve of the half maps and FSC curve for cross-validation between the map and model of unfolded CDTa class. **f, g** Density map of CDTa (folded or unfolded) and the surface model of subunit C and F of CDTb-pore. The map of CDTa was colored as N-terminal helices (red), N-terminal domain (yellow) and C-terminal domain (purple), respectively. The constrictions designated Ca-edge, NSS-loops and φ-clamp are shown in blue, cyan, and purple, respectively. **h, i** The model of the CDTa (folded or unfolded) N-terminus and the NSS-loops. The blue NSS-loops indicate 'in' state, and the yellow NSS-loops indicate 'out' state.

6KLO, respectively), in which we observed a weaker density for the 'out' conformation than for 'in' (Supplementary Fig. 6c). These results suggest that two loop conformations are common in the binary toxin family. The reasons for two conformations have not been addressed yet, but we speculate that flipping between the 'in' and 'out' states participates in translocation. In other words, flipping loops would mediate transient translocational states through non-specific interactions. Though the 'in' and 'out' conformations were seen in the short CDTa-bound CDTb-pore, 3DVA showed differences in the subunit D NSS-loop: 'in' and 'out' conformations could be seen in the unfolded

CDTa and the folded CDTa structure, respectively (Fig. 6 and Supplementary Fig. 7a).

From these analyses of the NSS-loop, we conclude that (1) 'out' is produced by steric hindrance between CDTa and the loop; (2) 'in' is produced by interactions such as hydrogen bonds (Supplementary Fig. 7b). The short CDTa-bound CDTb-pore map represents the total mixed map, including these small differences (Fig. 4a, b). Using SPR, we found that the more rigid mutant PSS-loop had a weaker interaction than either the WT NSS-loop or the mutant SPS-loop at pH 7.4 (in both high and low $Ca^{2+}$) and pH 5.5 (in low $Ca^{2+}$). Further

structural studies of CDTa translocation and cell toxicity are needed.

We focused on the relationships of the N-terminal α1 helix in the folded and the unfolded CDTa structures by 3DVA. The movie from the successive map confirms our image of protein threading into the pore (Supplementary Movie 1). We illustrate translocation in Fig. 7. First, CDTa binds to CDTb-pore, with α1 folded (Fig. 7b). Next, the entire CDTa molecule tilts and collides with CDTb (Fig. 7c). Finally, this triggers tilting and unfolding of the α1 α-helix (10-LKDKEKAKE-18) of CDTa. We previously reported the same region's unfolding (12-DKE-NAI-17) and tilting in the Ia-bound Ib-pore[37]. In both CDTa-bound CDTb-pore and Ia-bound Ib pore, the N-terminus of the A-component starts at almost the same position in the complex (18 in Ia and 19 in CDTa) (Fig. 5). Notably, in the unfolded CDTa structures, a similar density heading toward the φ-clamp was observed (Fig. 5) similar to that of the Ia-bound Ib-pore.

In summary, both pore complex structures allow limited space for the first α-helix due to steric hindrance between the substrate protein and the pore. The distance between the N-terminus of the first α-helix and the φ-clamp is 25–30-Å. We propose that the unstructured N-terminal regions of CDTa and Ia are essential for translocation via the φ-clamp. The first step of threading through the narrow φ-clamp is the most challenging, similar to threading a needle, with subsequent translocation proceeding more rapidly via the ΔpH-driven Brownian ratchet model[38]. For this purpose, the CDTb- and Ib-pore systems use a common mechanism to unfold the N-terminus of the substrate, induced by steric hindrance.

These unique binding and unfolding features also have been observed for both *C. difficile* toxin and *C. perfringens* iota toxin, suggesting that they are common in other binary toxins, including those of *C. spiroforme* and *C. botulinum* C2. The binding and unfolding of the A-components of CDT and iota toxin have been considered to be similar to that of anthrax toxin, but it was shown that CDT and iota toxin have a unique mechanism, which is different from anthrax toxin (Supplementary Note 1 and Supplementary Fig. 8).

Pharmacological inhibition studies to inhibit pore-forming toxins have started in CDT. A designed symmetrical cyclodextrin molecule has been shown to inhibit the CDTb-pore, as well as the PA-, Ib-, and C2-pores[39–41]. Furthermore, chloroquinone and its derivatives have been reported to be pore blockers[42]. Our high-resolution structures provide important insights for developing inhibitors to combat hypervirulent *C. difficile* infection.

## Methods

### CDTa and CDTb expression and purification

*cdtA* (Uniprot ID: Q9KH42, amino acids 51–463 (the numbering is 1-413 without signal peptide)) was cloned into pET-23a with a C-terminal TEV-protease recognition site followed by a His-tag and overexpressed in *Escherichia coli* C41. The transformants were cultured in 150 mL of LB medium containing ampicillin (50 µg/mL) and 2% (w/v) glucose at 37 °C for 16 h. Glucose was added to suppress the expression of CDTa at an early stage of the culture. The preculture medium was inoculated into 1.35 L of LB medium containing ampicillin (final concentration: 50 µg/mL) but no glucose so that the growing cells could express CDTa. The total volume (1.5 L) of LB medium was cooled in an ice-water bucket. Isopropyl β-D-1-thiogalactopyranoside (IPTG) was added (final concentration: 0.5 mM), followed by culturing at 23 °C for 5 h. The harvested cells were suspended in lysis buffer containing 20 mM Tris (pH 8.0) and 20 mM imidazole, disrupted by sonication in ice–water, and centrifuged at 180,000 × *g* for 40 min. The supernatant was

**Table 2 | Equilibrium binding affinity and kinetic parameters of the interaction between CDTa and CDTb**

| Buffer | Ligand | $k_a$ (×10⁵ M⁻¹s⁻¹) | $k_d$ (×10⁻⁴ s⁻¹) | $K_D$ (nM) |
|---|---|---|---|---|
| | WT CDTb | 1.44 | 41.3 | 28.6 |
| pH 7.4 | SPS | 1.16 | 30.4 | 26.2 |
| | PSS | 1.03 | 56.7 | 55.1 |
| | WT CDTb | 1.89 | 6.19 | 3.27 |
| pH 7.4, Ca²⁺ | SPS | 2.76 | 5.21 | 1.89 |
| | PSS | 2.2 | 10.8 | 4.93 |
| | WT CDTb | 4.98 | 7.63 | 1.53 |
| pH 5.5 | SPS | 4.41 | 7.09 | 1.61 |
| | PSS | 5.84 | 13.2 | 2.27 |
| | WT CDTb | 1.52 | 4.36 | 2.86 |
| pH 5.5, Ca²⁺ | SPS | 1.21 | 3.26 | 2.69 |
| | PSS | 1.64 | 3.76 | 2.29 |

All parameters were obtained in the analysis, which were shown in SI Fig. 4.

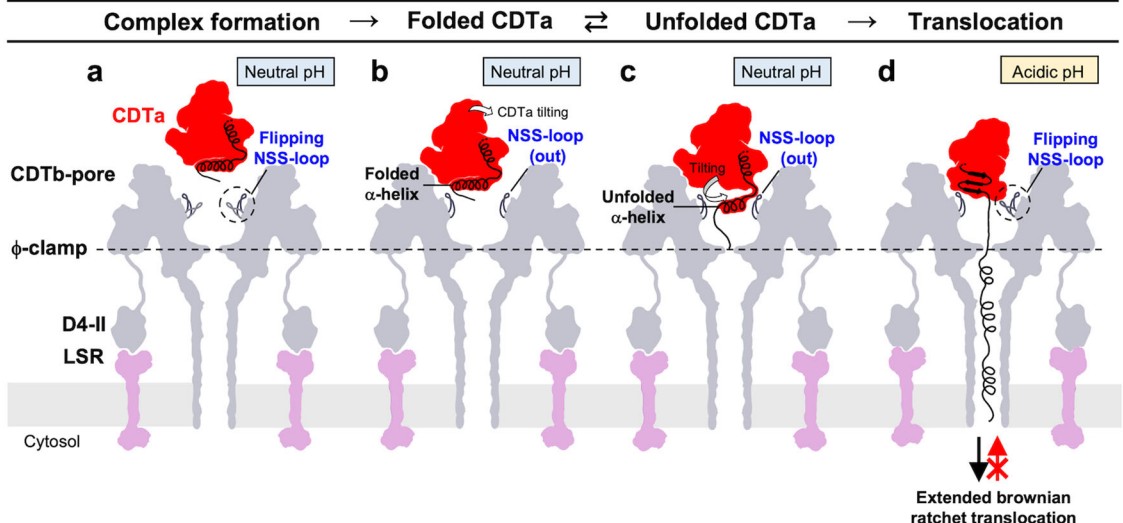

**Fig. 7 | Model for translocation of CDTa.** The illustrated protomers of CDTb-pore are equivalent to subunits C (left) and F (right). **a** CDTb-pore before CDTa binding. NSS-loops are shown in 'in' and 'out' states. **b** Single CDTa bound to CDTb-pore. **c** Unfolded CDTa also shown in Fig.6. CDTa tilting pushes the N-terminus of CDTa, causing partial unfolding and tilting of the N-terminal α-helix. **d** Unfolded CDTa translocates via the φ-clamp, using a ΔpH-driven Brownian ratchet mechanism, though the model is still an open question.

loaded onto a Ni-NTA agarose column. The column was washed with lysis buffer, and bound proteins, including CDTa, were eluted with a buffer containing 20 mM Tris (pH 8.0) and 500 mM imidazole. The eluted fractions were collected in an Amicon 30 K centrifugal tube to exchange the buffer with CDTa buffer containing 50 mM Tris (pH 8.0), 300 mM NaCl, and 2.5 mM CaCl₂. The C-terminal His-tag was removed by incubating the fractions with TEV-protease (1/10 weight of CDTa) at 37 °C for 1 h. Proteolysis was terminated by adding phenylmethylsulphonyl fluoride (PMSF) (final concentration: 1 mM). The CDTa was loaded onto SEC column Superdex 75 10/300 GL (Thermo Fisher Scientific) and eluted with a buffer containing 10 mM Tris (pH 8.0) and 100 mM NaCl at a flow rate of 0.5 mL/min. The fractions containing CDTa were collected and concentrated to 7.35 mg/mL using Amicon 30 K.

*cdtB* (Uniprot ID: o32739, amino acids 40–876) was cloned into pGEX4T-1 without the signal peptide and expressed in *E. coli* BL21 Star (DE3). The transformant was cultured in 1 L of super broth medium containing amipicillin (50 μg/mL) at 37 °C to an $OD_{600}$ of 0.7. The culture medium was cooled on an ice–water bucket. IPTG (final concentration: 1 mM) was added to the culture medium, and the culture was incubated at 20 °C for 16 h. The harvested cells were suspended in lysis buffer containing 20 mM Tris (pH 8.0), 150 mM NaCl, 2 mM CaCl₂, and 5 mM dithiothreitol and disrupted by sonication in ice–water. After centrifugation at 180,000 × *g* for 40 min, the supernatant was loaded onto a Glutathione Sepharose 4B resin (GE Healthcare) column. Then, the column was washed with lysis buffer and bound proteins, including CDTb, were eluted using a buffer containing 20 mM Tris (pH 8.0), 150 mM NaCl, and 10 mM reduced glutathione. The collected fractions were loaded onto the Amicon 50 K centrifugal tube to exchange the buffer containing 20 mM Tris (pH 8.0), 50 mM NaCl, and 2.5 mM CaCl₂, followed by a concentration to 1.1 mg/mL. The mutants of CDTb were prepared in the same protocol as wild type, with resulting concentration of F774L: 2.0 mg/mL, F774G: 2.0 mg/mL, PSS-loop: 3.1 mg/mL and SPS-loop: 2.0 mg/mL.

## Evaluation for formation of CDTb di-heptamer

To activate the oligomerization by removing the N-terminal 20-kDa pre-sequence and, 2 mg of wild-type CDTb was treated with 200 μg of α-chymotrypsin (SIGMA) for 1 h at 25 °C. Proteolysis was terminated by adding PMSF (final concentration: 1 mM). Then, the CDTb was incubated with LMNG (final concentration: 0.03% (w/v)) for 1 h at 37 °C, loaded onto a density gradient bed containing 10–30% (v/v) glycerol, 50 mM HEPES (pH 7.5), 100 mM NaCl, 1 mM CaCl₂ and 0.003% (w/v) LMNG, and ultracentrifuged at 230,139 × *g* for 16 h. Subsequently, the centrifuge tube was punctured at the bottom with an injection needle and five drops were collected as a fraction. The di-heptamer and heptamer distribution of CDTb-pores was monitored by measuring absorbance at 280 nm (A280). This series of experiments was termed "standard protocol".

To find out whether the timing of the addition of LMNG affects the formation of di-heptamer, the same amount of LMNG (final concentration: 0.03% (w/v)) was mixed with CDTb before treatment with α-chymotrypsin, but not after. The following experiment was performed totally the same as "standard protocol". To find out whether the mutation of F774 affects the formation of di-heptamer, using CDTb F774L or F774G, the experiments were performed with "standard protocol".

## Sample preparation for cryo-EM

The preparation of CDTb-pore for cryo-EM analysis, 1.1 mg of wild-type CDTb were treated with 1.1 μg of α-chymotrypsin for 1 h at 25 °C, followed by performing the "standard protocol" which is described above in the section of evaluation for formation of CDTb di-heptamer. After the density gradient ultracentrifugation, sodium dodecyl sulfate–polyacrylamide gel electrophoresis was performed, and

fractions containing CDTb-pores were collected. Buffer was exchanged with a buffer containing 10 mM HEPES (pH 7.5), 1 mM CaCl₂, and 0.003% (w/v) LMNG using PD10 and concentrated to 1.7 mg/mL. For complex formation, CDTa without His-tag was added to the resultant CDTb-pore at a 3-fold molar excess. Further purification was not performed to remove unbound CDTa.

## Cryo-EM data collection and image processing

The CDTa-bound CDTb-pore (1.58 mg/mL after the addition of CDTa) was applied to a glow-discharged Quantifoil holey carbon grid (R1.2/1.3, Cu, 300 mesh), blotted for 4.5 s at 4 °C in 100% humidity and plunged into frozen liquid ethane using a Vitrobot Mark IV (Thermo Fisher Scientific). The grid was inserted into a Titan Krios (Thermo Fisher Scientific) operating at an acceleration voltage of 300 kV and equipped with a Cs corrector (CEOS, GmbH). Cryo-EM images were recorded with a K3 direct electron detector (Gatan) in counting mode with an energy filter at a slit width of 20 eV. Data were automatically collected using the SerialEM software (https://bio3d.colorado.edu/SerialEM/) at a physical pixel size of 0.88 Å/pixel with a defocus range from −0.8 to −2.0 μm at 50 e/Å² with 3.36 s exposure.

The movie frames were subsequently aligned to correct for beam-induced movement and drift using MotionCor2[43], and contrast transfer function (CTF) was evaluated using Gctf[44]. Approximately 1000 particles were manually selected from 10 micrographs to perform two-dimensional (2D) classification. Using a good 2D class average image, a total of 2,608,418 particle images were automatically picked, and several rounds of 2D classifications were performed using RELION-3.1[45]. A total of 735,102 particles were selected for building the initial model of the CDTa-bound CDTb-pore using cryoSPARC2[46] and subjected to 3D classification into 10 classes using RELION-3.1, as shown in Supplementary Fig. 2. For the CDTa-bound CDTb-pore, a good 3D class was selected and subjected to 3D classification into three classes: di-heptamer, long stem and short stem. In long, the D4II domain of CDTb was clearly visible. By contrast, the D4II domain was not visible in short. The particles of long and short were re-extracted with a pixel size of 0.88 Å/pixel and subjected to three 3D refinements, two CTF refinements, and Bayesian polishing. Afterwards, 3D refinement and CTF refinement were repeated. For short subsets, no-align 3D classification using a mask focusing on the CDTa region improved the local resolution of CDTa. Final 3D refinement and post-processing yielded maps with global resolutions of 2.64-Å (long) and 2.56-Å (short), according to the 0.143 criterion of the FSC. Local resolution was estimated using RELION-3.1. The processing strategy is described in Supplementary Fig. 2.

Analysis of the di-heptamer of the CDTa-bound CDTb-pore was performed as follows: from the best class of the 3D classification containing 146,821 particles, duplicated particles were removed, and the remaining particles were re-extracted to a pixel size of 0.88 Å/pixel. A total of 102,116 particles were subjected to 3D refinement and CTF refinement. The final 3D refinement and post-processing yielded maps with global resolutions of 3.19-Å, according to the 0.143 criterion of the FSC. The processing strategy is described in Supplementary Fig. 2.

## Model building and refinement

**CDTa-bound CDTb-pore (long).** The model of the CDTa-bound CDTb-pore (long) was built using the cryo-EM density map of long pore. The initial rigid-body fit of the CDTa structure (PDB ID: 6V1S) and CDTb-pore structure (PDB ID: 6UWR) were applied to the map using UCSF Chimera[47]. The models of CDTa and CDTb-pore fitted into the map were saved individually. Next, single subunit of the heptameric CDTb-pore were manually modified and refined iteratively using COOT[48] and PHENIX real space refinement[49], respectively. The model of the tip of the β-barrel (337–358) was not built because of weak map density. Then, the single subunit of the heptameric CDTb-pore was symmetrically expanded and refined. An independently refined model of

CDTa was merged into the model of CDTb-pore. The map of NSS-loops revealed two conformations in some protomer. Thus, in subunits A, B, D, E and G, the residues (488–497) containing NSS-loop were constructed and refined as both "in" and "out" conformations with the occupancy of each conformation at 0.5:0.5, respectively. The long model was further manually modified and refined iteratively.

**CDTa-bound CDTb-pore (short).** The model of the CDTa-bound CDTb-pore (long) was fitted into the cryo-EM density map of short using UCSF Chimera rigid body fit. The tip of the β-barrel (332–336 and 359–363) and D4-II (743–876) were excluded because of poor map density. The model was manually modified and refined iteratively using COOT and PHENIX. In subunits A, B, D, E and G, the residues (488–497) containing NSS-loop were refined as both "in" and "out" conformations with the occupancy of each conformation at 0.5:0.5, respectively. They were further manually modified and refined iteratively using COOT and PHENIX.

### 3D variability analysis

From the short CDTa-bound CDTb-pore, 3D variability analysis was done using cryoSPARC[34,46]. At first, twenty class were obtained for the discrete conformational states of the short CDTa-bound CDTb-pore. A motion picture was obtained using a series of 20 maps, which represent from the folded to unfolded states of the N-terminal α-helix of CDTa in the complex (Supplementary Video 1). Then the 20 maps were divided into three classes followed by removing the duplicated particles; folded CDTa class (24,981 particles), intermediate class (37,141 particles) and unfolded CDTa class (100,912 particles). Each classes particles were further subjected to 3D refinement. The model of folded and unfolded CDTa-bound CDTb-pore were built using the templates CDTa-bound CDTb-pore (short) into the cryo-EM density maps, and folded CDTa was replaced with crystal structure of CDTa (PDBID: 2WN6) by UCSF Chimera rigid body fit. The N-terminal residues (1–18), which was not visible in the short map, were built in the folded CDTa. In the folded CDTa complex, the "in" conformations at subunit D was removed, because the density was not observed in the map. It was further manually modified and refined iteratively using COOT and PHENIX. On the other hand, in the unfolded CDTa complex, the "out" conformations at subunit D was removed, because the density was not observed in the map and it was further manually modified and refined iteratively using COOT and PHENIX.

For all structures, the flip states of the side-chain of Asn, Gln, and His residues were validated and corrected by protonation using MolProbity[50]. Gold-standard FSC curves of the final map, and FSC curves for cross-validation between the map and model, were evaluated using RELION-3.1 and comprehensive validation in PHENIX, respectively. All figures were prepared using UCSF Chimera, and UCSF ChimeraX[51].

### Surface plasmon resonance (SPR)

The sample of CDTb and NSS-loop mutants were prepared by using "standard protocol". Fractions containing CDTb-pores were collected. Buffer was exchanged with a buffer containing 10 mM HEPES (pH 7.5), 1 mM $CaCl_2$, and 0.003% (w/v) LMNG using PD10 followed by adjusting the concentration of WT (NSS-loop), PSS (PSS-loop) and SPS (SPS-loop) to 1.0 mg/mL, respectively. All SPR measurements were carried out using OpenPlex (HORIBA France). For the preparation of biochips, CDTb (WT (NSS-loop), PSS (PSS-loop mutant), SPS (SPS-loop mutant)) in PBS was immobilized onto an SPRi-Biochip CS-HD (HORIBA France) by the amine coupling method. CDTb (1.0 mg/mL) was spotted at using a spotter system (HORIBA, Ltd.), followed by reaction for 2 h at room temperature in a humid atmosphere. SPR experiments were performed at room temperature at a flow rate of 50 μL/min in following each conditions (A:pH 7.4 HEPES 10 mM 0.15 M NaCl 1 mM EGTA, 0.005% (v/v) Surfactant P20 B:pH 7.4 HEPES 10 mM 0.15 M NaCl 1 mM EGTA, 0.005% (v/v) SurfactantP20, $CaCl_2$ 5 mM C:pH 5.5 MES 10 mM 0.15 M NaCl 1 mM EGTA, 0.005% (v/v) Surfactant P20 D:pH 5.5 MES 10 mM 0.15 M NaCl 1 mM EGTA, 0.005% (v/v) Surfactant P20, $CaCl_2$ 5 mM). CDTa samples were injected, and the association of each solution was recorded for 180 s. Dissociation was monitored by injecting running buffer alone for 300 s, followed by injection of 100 mM glycine hydrochloride (pH 2.2) for 20 s in regeneration phase. Interactions between ligands and analyte on the surface shift the SPR curves. The change in reflectivity at a fixed angle was measured, and the signals were used to generate a kinetic curve. The reflectivity change (%) in the sample spots was calculated by subtracting the signals obtained at the untreated spots, where samples were not immobilized on the biochip surface from the signals recorded at the ligands spotted areas. Measured kinetic curves were analyzed using ScrubberGen software (HORIBA France). A 1:1 binding model was used to calculate $k_a$, $k_d$, and $K_D$ ($K_D = k_d/k_a$)[52]. Two measurements were taken from distinct samples, and the average values were shown.

### Reporting summary

Further information on research design is available in the Nature Research Reporting Summary linked to this article.

### Data availability

The Cryo-EM dataset was deposited to the Electron Microscopy Public Image Archive (https://www.ebi.ac.uk/pdbe/emdb/empiar/) under accession code of EMPIAR-11235. The Cryo-EM maps and coordinates were deposited to the Electron Microscopy Data Bank (EMDB) and Protein Data Bank (PDB) with the accession codes EMDB-33188 for CDTa-bound CDTb-pore di-heptamer, EMDB-32043 and PDB 7VNN for CDTa-bound CDTb-pore long, EMDB-32041 and PDB 7VNJ for CDTa-bound CDTb-pore short, EMDB-34136 and PDB 7YVQ for folded CDTa-bound CDTb-pore and EMDB-34137 and PDB 7YVS for unfolded CDTa-bound CDTb-pore, respectively.

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

## Acknowledgements

This work was supported by JSPS KAKENHI Grant Numbers 18K06170, 21H02452 (to H.T.) and 21J13410 (to T.Yamada), and JST PRESTO Grant Number JPMJPR21E5 (to A.K.). This research was partially supported by Platform Project for Supporting Drug Discovery and Life Science Research (Basis for Supporting Innovative Drug Discovery and Life Science Research (BINDS)) from AMED under Grant Number 2366. We thank S. Suzuki, H. Takahashi, S. Tomoda, and Y. Uchida of the members of Structural Biology Laboratory in Kyoto Sangyo University for help of the manuscript preparation.

## Author contributions

All author participated in research design; T.Yamada prepared the CDTb-pore and CDTa-bound CDTb-pore for cryo-EM; A.K. and T.K. performed cryo-EM data acquisition and A.K. and T.Yamada performed image processing and data analysis; T.Yamada and T.Yoshida performed the atomic model building, structure refinement and analyses; Y.S. performed the SPR data analyses; all author contributed to writing the manuscript and H.T. supervise the project.

## Competing interests

The authors declare no competing interests.
