## [Peer Review File · Nature Communications]

Cryo-EM structures of the translocational binary toxin complex CDTa-bound CDTb-pore from *Clostridioides difficile*REVIEWER COMMENTS

Reviewer #1 (Remarks to the Author):

In this work, Kawamoto et al describe the cryo-EM structure of the CDTa-bound CDTb pore at 2.56 Å resolution. The highest resolution of this structure showed that the first N-terminal alpha-helix of CDTa is tilted, with the first 10 amino acids adopting a dynamic structure (not observed). The structure also showed the dynamic conformation of the NSS-loop of CDTb, that may play a role in the transient grip of CDTa.

The authors describe a procedure to obtain heptamers using LMNG at 37°C. This procedure minimized the dimerization of the heptamers producing 3 classes of structures. The CDTa purified component was added to these LMNG-generated structures to generate CDTb pores complexes bound to CDTa. Two of the three observed structures corresponded to single CDTa-bound heptamer pores (class 1 and class 2).

Previously, Sheedlo et al (Ref 36) reported the structure of a CDTb-CDTa heptamer in the pre-insertion state, that was obtained in the presence of the receptor (LSR) ectodomain. A truncated form of CDTa (aa 17-420) with an N-terminal His6 and maltose-binding protein tags that were removed with PreScission protease, was used to obtain the structure of the pre-insertion heptamer with the CDTa subunit. Sheedlo et al reported a single CDTa subunit per heptamer at a resolution around 4 Å and described some of the residues involved in the interaction, including the location of the CDTa 1st alpha helix that showed a dynamic first 10 amino acids (not observed) of this truncated form.

The highest resolution obtained by Kawamoto et al allowed a more detailed description of the interaction of CDTa with the CDTb heptamer. The estimated surface area (1960 Å²) was larger than the previously described for the pre-insertion state (1517 Å²). First, the new structure provided details on the interaction of the NSS-loops and CDTa and the dynamics and grip that some of them had to interact with CDTa. Second, the location of the first alpha helix in CDTa show a tilt and the previously observed unfolded state of the N-terminus (however this helix was not truncated as in the previously described structure).

These observations are similar the ones previously described by the same Group for Iota toxin (Ref 41).

The manuscript is clearly written, and I did not find any problems with the methods and the provided supplemental material.

In summary, this work presents a very nice description of the CDTb-CDTa interaction for a monomeric heptamers (pore-state) at a higher resolution than previously. Details in these structures revealed the dynamic conformation (and potential role) of the NSS-loops for the binding of CDTa, previously missed in the electron density maps of previously resolved complexes. Also, it provided further details on the location of the N-terminal alpha-helix 1, that lead the authors to hypothesize a steric induced unfolding of the N-terminus of the helix, that may be required to initiate translocation. All the conclusions made in this work are from the observation and analysis of the obtained structure/s, no accompanied experimental data to support the proposed hypothesis was provided (not unusual for some structural research). Despite the highest resolution description of the complex, the slight differences found between the presented and previous published structures reduce the enthusiasm about the work.

Reviewer #2 (Remarks to the Author):

Recommendation: Publication after major revision

In the present work, A. Kawamoto, T. Yamada et al. describe the structure of the physiological form of the *Clostridioides difficile* binary CDT toxin, the heptameric CDTb pore in complex with one CDTa. The authors solved two $\sim 2.6\text{\AA}$ cryo-EM structures of the CDTa/CDTb(7) complex, which are differing in their completeness of the CDTb part (beta-barrel stem and receptor-binding domain D4II). In addition, the authors report a non-physiological CDTa/CDTb(7) -Dimer, in which the transmembrane regions of CDTb bind to each other. The latter has been reported previously without CDTa (Anderson et al, 2020; Xu et al, 2020), and also a CDTa/CDTb(7) pre-insertion complex (Sheedlo et al., 2020). The two physiological CDTa/CDTb(7) pore complexes, named class 1 and class2 by the authors, are the key novel finding of this work. The very high resolution of 2.6\AA allows to decipher a key step in CDTa binding and translocation: the flexibility of an identified NSS loop in CDTb that can exist in two distinct conformations (in/out); and in two out of seven CDTa-bound CDTb subunits the loop is fixed in the “out” conformation.

The same authors recently published structures of Iota toxin at $2.8 - 2.9\text{\AA}$ resolutions that show a very similar overall concept of translocation (Yamada et. al., NSMB, 2020). With the additional information obtained in the present work due to the so far unmatched resolution, the authors re-analyzed the Iota toxin structure and identified a similar conformational switch in the NSS loop like here. Therefore, the present work extends the conceptual understanding of this class of toxins and is suitable for publication in Nat Comm.

However, all interpretations in the manuscript are based on the structures. To proof the relevance of the NSS loop, functional or structural data of mutated CDTb NSS loops are necessary, such as rigidifying

the loop and locking in one conformation. Does this change the toxicity of CTD or does it change the CDTa/CDTb affinity?

Moreover, the authors build both loops in a 0.5:0.5 occupancy into the map. Are more focused refinements, sortings, eventually combined with signal subtraction possible to sort out specific conformations at the NSS loops? If the authors performed this without success, it should be stated in the manuscript.

Is a difference between the NSS loops in class 1 (with D4II bound to the beta-barrel stem) and in class 2 (D4II not resolved) eminent? Already outward-fixed NSS loops with stem contacts as depicted in Fig. 6, left panel, would be a disagreement between the scheme in Fig. 6 and the data.

I recommend to rename “classes 1 and 2” according to their features, e.g., class 1 as long-stem class. This would make reading more straightforward. The interpretation of CDTa binding and translocational unfolding dependent of the NSS loop switch is explained with class 2 only. What can additionally be interpreted with class 1? The unique findings in class1 are not described and interpreted – contacts between domain 4 with the beta-barrel stem, as depicted in the schematic in Fig. 6.

Further points to address:

Abstract:

The authors start with TcdA and TcdB, two *C. difficile* toxins that are not part of the work. I recommend to remove this distracting part and instead start with CDTa/CDTb as toxin translocation system.

I. 28: grammatical error (remove “a”)

Introduction:

I. 46f: “Additionally, some bacteria have a unique translocation system named binary toxin”: Can the system only translocate toxins or also other proteins? If the first is the case, I would define it as toxin translocation system.

I. 59: indicate a citation for “internalized by receptor-mediated endocytosis”.

I. 68 and 70: Is anthrax PA the “second” group? I recommend to write “second” instead of “another” in I. 68.

I. 74 - 86: I recommend to put the general chapter of *C. difficile* before the detailed description of binary toxin systems. The chapter here appears to be out of context.

I. 77: Are newer numbers than 2015 available?

I. 78: “certain strains”- are these strains responsible for *C. difficile* infection-related deaths?

I. 105: “fluctuation”- do you mean flexibility?

I. 110, classes 1 and 2: See above, I recommend to name them differently according to their unique features.

I. 112: Only class 2 is mentioned for interpretation – what about class 1?

I. 117: whose instead of which

Results:

I. 133 ff: As it is described here, LMNG was added after cleavage? Can LMNG during cleavage completely stop the formation of heptamer dimers, or eventually higher concentrations of the detergent? I recommend to show more data here as SI data. No biochemical and biophysical data at all are shown here (negative staining, SDS gels, ...).

I. 137/138: There is a sudden transition from sample preparation to cryo-EM 3D classification, a sentence describing electron microscopy would be helpful for understanding.

I. 140f: percentages instead of ratios of classes would be more intuitive.

I. 149: Write and interpret only one digit for cryo-EM structures.

I. 159: I recommend to re-phrase “seemed to contain one whole beta-barrel stem”, since it is not complete and therefore distracting.

I. 161: Is the entire proposed transmembrane part not resolved? Please indicate the TM part, also in Fig. 1.

I. 163: Not only class 2, but also class 1 do not contain the whole beta-barrel stem.

I. 165f: The presence of D4II is the most obvious difference in the two classes. Therefore, I recommend to elaborate and interpret this point further, possibly even in an own chapter in the results section. Under the view point of the activation and loading mechanism in Fig. 6, does the binding of D4II to the stem in the left image represent class 1?

I. 167: This sentence would be a better introductory sentence for the comparison of both classes. At the position here it seems misplaced.

I. 170ff, description of domains:

I. 177f: Which part of domain 4 binds to the receptor? How is the position here in class 1 in comparison to the receptor-bound state?

I. 180: Does the phi-clamp separate cis and trans? Then two constrictions would be on the cis side, while the phi-clamp separates.

I. 181: di-calcium site: Is there a difference in the seven subunits? As it is described in methods, no symmetry was applied. Are there Ca²⁺ binding differences with respect to CDTa binding?

I. 186: stable region – do you mean conserved region or physically stable?

I. 188: His 314 – please indicate in the seq alignment in SI Fig. 1 and describe exactly in which family the residue is conserved.

I. 192ff: Is the position of CDTa identical to the prepore state and to the position of Ia in Iota toxin? Are there other differences to the crystal structure than in the N-terminal helix? Is the N-terminal alpha helix artificially stabilized in the crystal structures of CDTa and Ia by contacts? If this is the case, no destabilization by CDTb would occur. If this is the case, please also check the corresponding section in the discussion.

I. 202ff, loop conformations: please indicate which residues were built in two conformations. Is the map/model fit and the geometric properties identically good in both conformations? Is the occupancy 0.5 and 0.5? Please also see my comment above (3D sorting, additional experiments).

I. 206: Please describe your re-analysis in the Methods section, and refer to SI figure 5 here, too. Since here C7 symmetry was applied, differences between subunits are not determinable. Does the 3.2 Å double pore structure obtained in this work show differences?

I. 210: “Thus, we conclude that the NSS-loop conformation is in equilibrium between the two states in the default pore structure” – in the sentences above, it is described that both conformations exist both in the prepore and in the pore part. Please double-check and re-phrase unambiguously.

I. 214ff: I recommend to introduce a new chapter heading for CDTa. How “deeply” does CDTa penetrate into the pores?

I. 219: Is density at low binarization threshold and/or after lowpass filtering visible that corresponds to the missing residues? In Methods, it is described that CDTa starts at residue 53 – please explain this discrepancy and re-number residues.

I. 220: Is the “steric hindrance” part of the mechanism? Here, it can be interpreted that it is non-physiological. From an unbiased point of view, I would expect that the missing residues are stabilized by the contacts with CDTb. Are data available that an N-terminally truncated CDTa variant binds to CDTb and/or is translocated? If not, I suggest an additional experiment to confirm this.

Discussion:

I. 233: Does a mutation of F774L in CDTb also disfavor heptamer dimerization? If no data are available, this would be a quick experiment to perform.

I. 237f: Although the di-heptamer might be protected from proteolysis, does it have any physiological relevance? Is di-heptamerization reversible? Since the authors suggest a concentration-dependent behavior, is it possible to confirm this and find out the K(D)?

I. 247: This would make the NSS loop to a NSX loop.

I. 249: use “a” instead of “the”

I. 257f: Please describe in the methods the model re-building of Ib. Can the “weaker” out-conformation density than in-conformation density be described in a quantitative manner? In addition, the fact that

two different loops are locked in the out state in Ib is not discussed at all. What are possible causes or consequences?

I. 261: “not so strong” – how strong does the binding need to be? I would estimate that the $K(d)$ has to be in the range of physiological concentrations. Are there affinity constants available for CDT or Iota or other, comparable toxins of this family?

I. 265ff, anthrax PA: A SI figure that compares the dimensions of CDT/Iota and anthrax PA would be helpful.

I. 275: Where is the alpha clamp in CDTb and Iota, this has not been stated in the manuscript before. Is it the C-alpha edge?

I. 280: the N-terminal helix of CDTa and Ia?

I. 314f: The open states were only observed for Anthrax PA, which has likely another mechanism. Are any indications for a phi-clamp opening available in CDT or Iota toxins?

I. 317: What did the cited study of the PA translocation complex show? Is it relevant for the CDT translocation mechanism?

I. 322: Can in silico docking experiments reveal how the described inhibitors could block translocation?

Methods: Some issues (missing description of re-modelling of two structures) were already described in the corresponding results section.

Sample preparation: Can LMNG already be added for proteolysis? Does this inhibit heptamer dimerization completely (or higher conc. Of LMNG)? Has unbound CDTa been removed for cryo-EM?

Cryo-EM data collection: At which concentration was the protein vitrified?

Data processing: Has C7 symmetry been applied for any step of data processing? Please also show the initial model of cryosparc that was used for the initial 3D sorting in SI Fig. 2. At which pixel size were the initial steps carried out (until re-extraction)?

I.404: Alternating CTF refinements and 3D refinements? What resolution improvement did this step yield?

I.406: Please describe the “non-align 3D classification” of CDTa in more detail – were all particles used for this part afterwards? Were misoriented particles rotated into an orientation that CDTa fits?

I. 419: The 2.9 Å resolution is not shown in SI Fig. 2.

Model building and refinement: Were both classes 1 and 2 used equally for building CDTa, which is much better resolved in class 2? Were all seven CDTb models built independently in both classes?

I. 445: Show map-to-model FSCs in SI Fig. 2!

Figures: Some figures show redundant data and representations. As an example, Figures 3 and 5 could be merged, since panels A and B do not show more than Fig. 3 a,b.

Fig.1: Indicate the transmembrane part at the maps (possibly class1 only). Consider merging figures 1 and 2.

Fig. 2c,d: Are the CDTa models in classes 1 and 2 identical?

Fig.3: Here, additional panels with surface representation of the NSS loops (hydrophobicity, charge) could reveal more insights into transient binding of CDTa. Panel a: show the dimensions of the circles. Panel c: "side views" instead of close-up views. Panel d: Show additionally how the in-conformations of the NSS loop would interact with CTDa. Does this reveal why the conformations are not present in the two subunits? L. 527: typo.

Fig. 4: A comparison of the crystal structure directly with the cryo-Em structure would be more helpful in panel a instead of the per-structure alignment of CDTa and Ia. This is also redundant with SI fig. 1b. c,d: show tilting angle. Define colors for crystal and EM structure in legend.

Fig. 5: This figure could be merged with Fig. 3, and iota toxin could be shown in the same way like in Fig. 3b. Or, alternatively, show all NSS loop in CDTb, Iota, in prepores and pores, in one SI figure (analogous to SI Fig. 4 and 5). Name pdb IDs of Iota toxin. The in-conformation is blue, not purple (l. 543).

Fig. 6: Translocational model could be extended by pH values and assembly of holotoxin, I would recommend to extend the mechanism according to Fig. 6 in the NSMB 2020 paper of the authors.

Table 1: Give preliminary pdb and EMDB values, and also show the dimer map for completeness.

Supplementary data:

SI 1: Typo in legend; indicate the position of His 314, which is mentioned in the text. Are the 2ndary structures in b from the crystal structures or from the structure here?

SI2: Please add map-to-model FSCs, the initial cryosparc model, and more details of the Focused 3D classification.

SI Fig. 3 could be merged with SI Fig. 2.

SI Fig. 5: A side-by-side comparison of all CDTb loops and Iota loops in all investigated conformations would be helpful, possibly merge with SI Fig. 4. Please also state which ones were processed with C7 symmetry.

Reviewer #3 (Remarks to the Author):

In their manuscript "Cryo-EM structures of the translocational binary toxin complex CDTa-bound CDTb-pore from *Clostridioides difficile*", Kawamoto and coworkers present a structure model for the protein complex formed by the two components of the binary toxin CDT from *Clostridioides difficile*, which is based on detailed cryo-EM analysis. Moreover, they provide a more detailed model for the binding of the enzyme subunit CDTa to the heptameric binding subunit CDTb and the insertion and translocation of partially unfolded CDTa into and through the CDTb pore.

The results of this study confirm and extend the knowledge about the structure and function of CDTa/CDTb, which was obtained in recent years by other groups, also using cryo-EM as well as further biophysical approaches such as X-ray crystallography and NMR. These former studies revealed a CDTa/CDTb complex with a heptameric structure for CDTb, too, and they also proposed a model for the interaction of CDTb with CDTa and the translocation of CDTa via CDTb. However, in these earlier studies, CDTb was found to be a di-heptamer, while the present work, most likely due to its novel purification protocol for the CDTb pore and a higher resolution, demonstrates that in addition to di-heptamers, CDTb also exists as a single heptamer in the CDTa-bound CDTb pores. This is a central new finding of this manuscript. The authors suggest in their discussion that this is the biologically active CDT complex, which exists in low CDTb concentrations and exhibits toxic effects on cells by delivering CDTa, while CDTb di-heptamers might exist under high CDTb concentrations to prevent aggregation and/or degradation of this protein.

Based on cryo-EM, the authors performed a detailed analysis of CDTb heptamers and their interaction with CDTa. They found four crucial elements of the CDTb heptamers, i.e. calcium edges that serve for binding of CDTa, NSS-loops that are also involved in CDTa binding and likely crucial for CDTa unfolding and insertion into CDTb pores, a Phe-clamp, which was earlier found in PA63 and Ib heptamers from the related binary anthrax and iota toxins, too, and a stem which forms the channel below the Phe-clamp.

Their results revealed that CDTa has a N-terminal and C-terminal domain and the first two alpha-helices of the N-terminal domain partially unfold after binding to CDTb pores and 9 amino acid residues within the first helix penetrate deeply into the pore and induce the unfolding of CDTa. Here, the NSS loops of CDTb are involved by switching between an "in" and "out" conformation, which makes the strong binding of CDTa to CDTb more loosely and enables translocation into the channel of CDTb pores.

Here, it would be very interesting to investigate the CDTb/CDTa interaction under acidic conditions, as in endosomes where CDTa translocation through CDTb pores occurs in living cells. This could be included in the manuscript in direct comparison to the interaction under neutral pH conditions.

We thank three reviewers for their careful reading and helpful comments. Please find below our point-by-point response. In the revised manuscript, we added two new biochemical data: One is mutational studies of F774 (receptor binding domain's key residue) to see how important this residue is for the di-heptamer formation. The second one is the CDTa-CDTb-pore binding studies of wt and two mutants of NSS-loop (NSS>SPS and NSS>PSS) by surface plasmon laser. Furthermore, a new 3D variability analysis of short CDTa-bound CDTb-pore showed CDTa movements from a folded to an unfolded state. We hope that these new data strengthen our findings and the paper will be acceptable in Nature communications.

Reviewer #1

In summary, this work presents a very nice description of the CDTb-CDTa interaction for a monomeric heptamers (pore-state) at a higher resolution than previously. Details in these structures revealed the dynamic conformation (and potential role) of the NSS-loops for the binding of CDTa, previously missed in the electron density maps of previously resolved complexes. Also, it provided further details on the location of the N-terminal alpha-helix 1, that lead the authors to hypothesize a steric induced unfolding of the N-terminus of the helix, that may be required to initiate translocation. All the conclusions made in this work are from the observation and analysis of the obtained structure/s, no accompanied experimental data to support the proposed hypothesis was provided (not unusual for some structural research). Despite the highest resolution description of the complex, the slight differences found between the presented and previous published structures reduce the enthusiasm about the work.

>

Thank you for your comments. We provided further details of N-terminal α -helix 1, and this discussion was further studied by 3D variability analysis, which presents CDTa motion in the complex. We added the motion picture of the N-terminal α -helix unfolding to thread it to the ϕ -clamp (Supplementary Video). Furthermore, we added two new biochemical data: One is mutational studies of F774 (receptor binding domain's key residue) to see how important this residue is for the di-heptamer formation (new Fig.1). The second one is the CDTa-CDTb-pore binding studies of wt and two mutants of NSS-loop (NSS>SPS and NSS>PSS) by surface plasmon laser (new Table 2 and new Fig. S4). We hope these data support the first details of the unfolding mechanism of CDTa via CDTb-pore.

Reviewer #2

With the additional information obtained in the present work due to the so far unmatched resolution, the authors re-analyzed the Iota toxin structure and identified a similar conformational switch in the NSS loop like here. Therefore, the present work extends the conceptual understanding of this class of toxins and is suitable for publication in Nat Comm.

>

Thank you for your valuable comments. We sought to respond as much as we can and provide two additional biochemical data.

(1)

However, all interpretations in the manuscript are based on the structures. To proof the relevance of the NSS loop, functional or structural data of mutated CDTb NSS loops are necessary, such as rigidifying the loop and locking in one conformation. Does this change the toxicity of CTD or does it change the CDTa/CDTb affinity?

> We added new SPR data in the last result section.

Considering the importance of the NSS-loop for CDTa binding and translocation, we sought to determine whether NSS-loop rigidity affects the CDTa binding using surface plasmon resonance (SPR).

~

Though PSS rigidity did not cause a major change in CDTa-CDTb binding, we speculate that its rigidity should affect translocation. (L244~260)

(2)

Moreover, the authors build both loops in a 0.5:0.5 occupancy into the map. Are more focused refinements, sortings, eventually combined with signal subtraction possible to sort out specific conformations at the NSS loops? If the authors performed this without success, it should be stated in the manuscript.

> We tried the more focused refinements, but it did not succeed in differentiating the two states (in and out states of NSS-loop) clearly. To see the variability, we tried the 3D variability analysis using cryoSPARC2. In the input, we feed all particles from relion's analysis of short CDTa-CDTb-pore (new Fig.S2 and new Fig.6). Then, using a series of maps, we generated one movie (Supplementary Video) and added two refined structures, the folded CDTa complex and the unfolded CDTa complex. Though we could not evaluate the accurate occupancy of NSS-loop, we found

differences of the NSS-loop conformations upon the folded and the unfolded CDTa complex by the 3DVA analysis. We added as follows " Two NSS-loop conformations would adapt to fit transient translocational CDTa states: the unfolded and folded complexes showed different 'in' and 'out' conformations in the subunit D NSS-loop (Fig. 6h,i). The short CDTa-bound CDTb-pore map represents the total map, showing the small differences between folded and unfolded CDTa (Fig. 4a,b)." (L239-243)

We believe these are very exciting data to support our discussion of the initial unfolding process of CDTa via CDTb.

(3-1)

Is a difference between the NSS loops in class 1 (with D4II bound to the beta-barrel stem) and in class 2 (D4II not resolved) eminent? Already outward-fixed NSS loops with stem contacts as depicted in Fig. 6, left panel, would be a disagreement between the scheme in Fig. 6 and the data. I recommend to rename "classes 1 and 2" according to their features, e.g., class 1 as long-stem class. This would make reading more straightforward. The interpretation of CDTa binding and translocational unfolding dependent of the NSS loop switch is explained with class 2 only. What can additionally be interpreted with class 1? The unique findings in class1 are not described and interpreted – contacts between domain 4 with the beta-barrel stem, as depicted in the schematic in Fig. 6.

>

As the reviewer suggestion, we renamed long (class 1) and short (class 2). Maybe old Fig. 6 leads to the misunderstanding which there is any correlation between stem formation and two conformations of NSS-loop. Sorry for this. We revised new Fig. 7 focusing on CDTa binding and CDTa translocational unfolding.

The interpretation of CDTa binding and translocational unfolding dependent of the NSS loop switch is explained with class 2 only.

>

In the long map, we also observed not only the unfolding and tilting of the first N-terminal α -helix of CDTa but also the same two states of NSS-loop as observed in the short map. However, the short map quality is better than the long, especially in CDTa. Thus, we went in the next step by the 3DVA using in the short map.

The unique findings in class1(long) are not described and interpreted

>

We added the relationship between domain D4II and the stem formation in the second paragraph of discussion. "The long and short structures, with other reported structures, of CDTb, including di-heptamers, are summarised in Supplementary Fig. S5. Between the long and the short structures, the definitive difference is that D4II is clearly visible in the long form. That is, after stem formation, D4II is stably fixed around the stem. In di-heptamer structures, two different conformations of D4II were observed. These dynamic features of D4II are likely important in binding the host cell LSR. In our previous study, we confirmed that the receptor-binding domain is also involved in determining oligomerization efficiency, using a D4II-deficient mutant Ib¹."

(L271~278)

(4)

Further points to address:

Abstract:

The authors start with TcdA and TcdB, two *C. difficile* toxins that are not part of the work. I recommend to remove this distracting part and instead start with CDTa/CDTb as toxin translocation system.

l. 28: grammatical error (remove "a")

>Thank you for your comment. I revised the first part of the abstract.

(we deleted the TcdA and TcdB parts from abstract.)

(5)

l. 68 and 70: Is anthrax PA the "second" group? I recommend to write "second"

instead of "another" in l. 68.

> I revised it.

(6)

l. 74 - 86: I recommend to put the general chapter of *C. difficile* before the detailed description of binary toxin systems. The chapter here appears to be out of context.

> I revised it.

(7)

l. 77: Are newer numbers than 2015 available?

We refer to "Antibiotic Resistance Threats in the United States, 2019-CDC"

<https://www.cdc.gov/drugresistance/biggest-threats.html>

(8)

l. 78: "certain strains"- are these strains responsible for C. difficile infection-related deaths?

> Yes, CdtB, the pore-forming component of CDT were more likely to have severe disease with higher mortality in Hypervirulent strains such as ribotype 027.

(9)

l. 105: "fluctuation"- do you mean flexibility?

> Yes, we revised to flexibility.

(10)

l. 110, classes 1 and 2: See above, I recommend to name them differently according to their unique features.

> We renamed long and short.

(11)

l. 112: Only class 2(short) is mentioned for interpretation – what about class 1(long)?

> Please see the above comment of 3-1.

(12)

l. 117: whose instead of which

> Corrected

(13)

Results:

l. 133 ff: As it is described here, LMNG was added after cleavage?

>Yes

Can LMNG during cleavage completely stop the formation of heptamer dimers, or eventually higher concentrations of the detergent? I recommend to show more data here as SI data. No biochemical and biophysical data at all are shown here.

> We added new biochemical and biophysical experiments of di-heptamer formation (Fig.1a). There was no difference of di-heptamers when LMNG added before chymotrypsin treatment (Fig 1b). There was also no difference of the ratio of di-heptamer and heptamer formation when 5 times LMNG added (data not shown).

Furthermore, we showed new data of F774L and F774G: F774 is the critical residue for the di-heptamer formation by the density gradient centrifugation method (Fig 1e and f).

(14)

l. 137/138: There is a sudden transition from sample preparation to cryo-EM 3D classification, a sentence describing electron microscopy would be helpful for understanding.

> We added a sentence describing the electron microscopy. L143 ~147

(15)

l. 140f: percentages instead of ratios of classes would be more intuitive.

> Corrected. "The ratios of the di-heptamer, long pore, and short pore were approximately 29:23:48%, with particle numbers 102116, 83061, and 173260, respectively.)" L148~150

(16)

l. 149: Write and interpret only one digit for cryo-EM structures.

>

Long and short at 2.64- and 2.56-Å resolution, respectively.

Because I would like to show that the short is higher resolution structure (substantially, the map is better than long in CDTa part), thus I would like to keep it.

(17)

l. 159: I recommend to re-phrase "seemed to contain one whole beta-barrel stem", since it is not complete and therefore distracting.

> OK. We changed as follows.

"The former contained one entire β -barrel stem; however, the final map calculation showed that the tip of this stem (residues 337–358) had very weak density; therefore, we modelled the structure with a partial β -barrel stem, excluding residues 337–358 (Supplementary Fig.S2d)." L167~170

(18)

l. 161: Is the entire proposed transmembrane part not resolved? Please indicate the TM part, also in Fig. 1.

l. 163: Not only class 2, but also class 1 do not contain the whole beta-barrel stem.

>As we described in the method, the long have some visible map of whole stem. However, it is not possible to put the whole stem model because of the weak density. We added the TM part in Fig.2. Also, we showed the difference of the stem between long and short in Supplementary Fig. S2d.

(19)

l. 165f: The presence of D4II is the most obvious difference in the two classes. Therefore, I recommend to elaborate and interpret this point further, possibly even in an own chapter in the results section. Under the view point of the activation and loading mechanism in Fig. 6, does the binding of D4II to the stem in the left image represent class 1?

>

Yes.

Thank you for your suggestion. We added the relationship between D4II domain and the stem formation in the discussion including the former results (L271-278).

However, we deleted the relationship between D4II domain and the stem formation from new Fig.7, because we think that it confuses about the two conformations of the NSS-loop. We also revised the new Fig.7, focusing on the NSS-loop and CDTa motions (see 3DVA analysis).

(20)

l. 167: This sentence would be a better introductory sentence for the comparison of both classes. At the position here it seems misplaced.

>

I wrote it in the introduction " In both structures, one CDTa molecule binds to the heptameric CDTb subunit through its N-terminal domain. "(L101~102)

However, the precise structures were explained here with Fig.3ab.

(21)

l. 170ff, description of domains:

> These are description of domains as described in Fig.3ab.

(22)

l. 177f: Which part of domain 4 binds to the receptor?

> Domain 4 binds to the receptor via F774, which was confirmed in Lacy's paper².

Also, please see the new Fig1d.

How is the position here in class 1 in comparison to the receptor-bound state?

>

The receptor -bound states were revealed as CDTa-bound CDTb-preinsertion state at 3.8Å resolution³. However, LSR receptor and D4II is missing due to their flexibility. Thus, we can not evaluate the F774 position in the receptor-bound state.

(23)

l. 180: Does the phi-clamp separate cis and trans? Yes

Then two constrictions would be on the cis side, while the phi-clamp separates.

>Yes. We deleted one sentence and just explained as follows and showed in Fig. 3e and 3f. " The lumen of the pore contained four constrictions. "(L188)

(24)

l. 181: di-calcium site: Is there a difference in the seven subunits?

>No. As it is described in methods, no symmetry was applied.

Are there Ca²⁺ binding differences with respect to CDTa binding?

> There are no differences in the di-calcium site.

(25)

l. 186: stable region – do you mean conserved region or physically stable?

Phi-clamp side chain is clearly visible, so I think it is physically stable.

(26)

l. 188: His 314 – please indicate in the seq alignment in SI Fig. 1 and describe exactly in which family the residue is conserved.

> We added as "H314 had a density of two conformers and was conserved in the family (CDTb, Ib and CSTb)."

(27)

l. 192ff: Is the position of CDTa identical to the prepore state and to the position of Ia in Iota toxin?

>Yes, CDTa locates identical position compared with Ia.

Are there other differences to the crystal structure than in the N-terminal helix?

> No.

Is the N-terminal alpha helix artificially stabilized in the crystal structures of CDTa and Ia by contacts?

>No, several crystal structures (CDTa and Ia) were solved in different space groups, and their N-terminal alpha helix were kept in the same conformations.

(28)

l. 202ff, loop conformations: please indicate which residues were built in two conformations.

In method L502 " Thus, in subunits A, B, D, E and G, the residues (488–497) containing NSS-loop were constructed and refined as both “in” and “out” conformations with the occupancy of each conformation at 0.5:0.5, respectively. "

Is the map/model fit and the geometric properties identically good in both conformations? >Yes

Is the occupancy 0.5 and 0.5? Yes Please also see my comment above

>I replied in (2).

(29)

l. 206: Please describe your re-analysis in the Methods section, and refer to SI figure 5 here, too.

> Though we did not refine the coordinate, we confirmed two states in the di-heptamer by visual inspection of the map. We added in the legend of Fig.S6 as follows. "CDTb di-heptamer cryo-EM map (EMD-20926) and Ib-pore map (EMD-0721) were visually inspected for conformations of NSX-loops. "

Since here C7 symmetry was applied, differences between subunits are not determinable. Does the 3.2 Å double pore structure obtained in this work show differences?

> We obtained the maps of CDTa-bound di-heptamer. Even we applied C1 symmetry calculation, CDTa and the binding site in CDTb were averaged. In this meaning, we did not model of the structure.

(30)

l. 210: "Thus, we conclude that the NSS-loop conformation is in equilibrium between the two states in the default pore structure" – in the sentences above, it is described that both conformations exist both in the prepore and in the pore part. Please double-check and re-phrase unambiguously.

> Thank you. As shown in the map new SI fig. 6a and b, both conformations exist both in the prepore and in the pore part. Thus, we revised as follows "Thus, we conclude that the NSS-loop conformation is in equilibrium between states in the default pre-pore and pore structures." L218~220

(31)

l. 214ff: I recommend to introduce a new chapter heading for CDTa.

> I changed the name of the chapter "**CDTa-binding mode and translocational unfolding**".

How "deeply" does CDTa penetrate into the pores?

> I deleted "deeply".

(32)

l. 219: Is density at low binarization threshold and/or after lowpass filtering visible that corresponds to the missing residues?

We added new results from 3D variability analysis of the short CDTa-bound CDTb-pore and refined two coordinates the folded CDTa and the unfolded CDTa in the complex. In the unfolded CDTa, we confirmed the density heading to the ϕ -clamp, caused by N-terminal α -helix unfolding. We added this in the results and discussion

of 3D variability analysis. (L230~243 and L321-331)

(33)

In Methods, it is described that CDTa starts at residue 51 – please explain this discrepancy and re-number residues.

> It includes signal peptide. In Ia case, the signal peptide was shown in Popoff paper Fig.3⁴. Thus, CDTa numbering was referenced to the Ia sequence.

We revised as follows " *cdtA* (Uniprot ID: Q9KH42, amino acids 51–463 (the numbering is 1-413 without signal peptide)) was cloned into pET-23a with a C-terminal TEV-protease recognition site " (L355~356)

(34)

l. 220: Is the "steric hindrance" part of the mechanism? Here, it can be interpreted that it is non-physiological.

> I think it is indeed physiological.

From an unbiased point of view, I would expect that the missing residues are stabilized by the contacts with CDTb. Are data available that an N-terminally truncated CDTa variant binds to CDTb and/or is translocated? If not, I suggest an additional experiment to confirm this.

> In sheedle paper(PNAS 2020) Fig.3, they have already confirmed that an N-terminally truncated CDTa variant (delta alpha1- alpha 4) did not show toxicity.

(35)

Discussion:

l. 233: Does a mutation of F774L in CDTb also disfavor heptamer dimerization? If no data are available, this would be a quick experiment to perform.

>Please see the above comments in (13). We added new data of F774L and F774G.

(36)

l. 237f: Although the di-heptamer might be protected from proteolysis, does it have any physiological relevance? Is di-heptamerization reversible? Since the authors suggest a concentration-dependent behavior, is it possible to confirm this and find out the K(D)?

>

We find that di-heptamer and heptamer could be separated by cryo-EM classification and also by ultracentrifugation. We tried blue-native page, but it results that di-heptamer break to single heptamer (data not shown). From this, we could not evaluate whether it is a concentration-dependent behavior.

Though it is not sure that the di-heptamer is physiologically important, it is sure that di-heptamer can not stick into lipid bilayer. I added the results as follows " When WT is bound to LSR via F774², it should be noted that it competes with the formation of a non-functional di-heptamer." (L137~138)

(37)

l. 247: This would make the NSS loop to a NSX loop.

> I changed to NSX-loop.

(38)

l. 249: use "a" instead of "the"

> Corrected.

(39)

l. 257f: Please describe in the methods the model re-building of Ib.

>

We added in the legend of Fig.S6 as follows. "CDTb di-heptamer cryo-EM map (EMD-20926) and Ib-pore map (EMD-0721) were visually inspected for conformations of NSX-loops. "

Can the "weaker" out-conformation density than in-conformation density be described in a quantitative manner?

>No. But there are clear difference between in and out of visible density.

In addition, the fact that two different loops are locked in the out state in Ib is not discussed at all. What are possible causes or consequences?

>

We can provide clear explanation of CDTb in the discussion with 3DVA analysis (Supplementary Fig. S7). In Ib case, we could explain subunit F : "out" is caused by steric hindrance like CDTb. However, in subunit E, we could not explain well at present.

(40)

l. 261: "not so strong" – how strong does the binding need to be? I would estimate that the $K(d)$ has to be in the range of physiological concentrations. Are there affinity constants available for CDT or Iota or other, comparable toxins of this family? CDTa-CDTb.

> We added the new table 2 for K_D (the dissociation constant) of wt and two NSS-loop mutants. Also, we deleted the word "strong". We revised as follows " The reasons for two conformations have not been addressed yet, but we speculate that flipping between the 'in' and 'out' states participates in translocation. " (L305~307)

(41)

l. 265ff, anthrax PA: A SI figure that compares the dimensions of CDT/Iota and anthrax PA would be helpful.

>

The discussion of the comparison with PA was added as supplementary note because it is better to focus on CDTb (and Ib) in the discussion section. We also added new figure for the comparison with CDT, Iota, and PA with the dimensions (new Supp Fig.S8).

(42)

l. 275: Where is the alpha clamp in CDTb and Iota, this has not been stated in the manuscript before. Is it the C-alpha edge?

>

We showed the region of alpha clamp of PA in the new Supp Fig.S8 figure. There is the similar regions in CDTb and Iota, but the site does not function to bind α -helix.

(43)

l. 280: the N-terminal helix of CDTa and Ia?

> We deleted this sentence of the comparison with PA. Instead of this, we added new discussion of 3DVA.

(44)

l. 314f: The open states were only observed for Anthrax PA, which has likely another mechanism. Are any indications for a phi-clamp opening available in CDT or Iota toxins?

>There are no other indications for CDTb and Ib.

(45)

l. 317(316?): What did the cited study of the PA translocation complex show? Is it relevant for the CDT translocation mechanism?

>They showed the first translocation model based the cryo-EM structure⁵. We do not know the PA translocation model is universal for Iota group binary toxin, thus we sought to catch the transient complex structure of CDTb or Ib. (At present version, we deleted this section. Please see the comment (42).)

(46)

l. 322: Can in silico docking experiments reveal how the described inhibitors could block translocation?

>This is next theme for iota and CDT. We are now working with in silico studying group.

(47)

Methods: Some issues (missing description of re-modelling of two structures) were already described in the corresponding results section.

>Please see the comment (29).

Sample preparation:

Can LMNG already be added for proteolysis?

> No, we do not add for proteolysis. (This is standard protocol as described in new Fig.1a) Furthermore, we checked whether the timing of LMNG (before and after protease treatment) would affect the di-heptamer or heptamer formations (new Fig.1b,c). But we did not observe any difference of the di-heptamer or heptamer formations. We also showed the two mutants F774L and F774G decreased the di-heptamer fraction (new Fig.1e and f). (Please see the comment (13).)

Does this inhibit heptamer dimerization completely (or higher conc. Of LMNG)?

>No, we did not inhibit the di-heptamer formation with five times more LMNG (data not shown).

Has unbound CDTa been removed for cryo-EM?

>No, we did not .

This is the same protocol as iota-toxin Ia-bound Ib-pore structural determination⁶.

Thus, we added the method.

(48)

Cryo-EM data collection: At which concentration was the protein vitrified?

>We added the method. (L428)

Data processing: Has C7 symmetry been applied for any step of data processing?

>No, we did not.

Please also show the initial model of cryosparc that was used for the initial 3D sorting in SI Fig. 2. At which pixel size were the initial steps carried out (until re-extraction)?

>In new Supp Fig.S2, we added pixel size in each steps.

1.404: Alternating CTF refinements and 3D refinements? What resolution improvement did this step yield?

>We added the resolution improvement of each steps in new Supp Fig.S2.

1.406: Please describe the "non-align 3D classification" of CDTa in more detail – were all particles used for this part afterwards? Were misoriented particles rotated into an orientation that CDTa fits?

>We added the information (mask and outputs) about the non-align 3D classification in new SI Fig. 2.

1. 419: The 2.9 Å resolution is not shown in SI Fig. 2.

> Sorry for this. We deleted this part of SI Fig. 2 and the method.

(49)

Model building and refinement: Were both classes 1 and 2 used equally for building CDTa, which is much better resolved in class 2?

>

We revised the method. At first, we built the CDTa-bound CDTb-pore (long) and next we built the CDTa-bound CDTb-pore (short), independently. However, CDTa was much better resolved in short map. Furthermore, from 3DVA of the CDTa-bound CDTb-pore (short), we refined two more coordinates for folded CDTa and unfolded CDTa complex. In both folded CDTa and unfolded CDTa complex map, the map quality of CDTa are

excellent, as seen in new Fig.6b and c (and Supplementary Video).

(50)

l. 445: Show map-to-model FSCs in SI Fig. 2!

> Yes, we added.

(51)

Figures: Some figures show redundant data and representations. As an example, Figures 3 and 5 could be merged, since panels A and B do not show more than Fig. 3 a,b.

> We reconsidered all figures including new figures (new figures are Fig.1, Fig.6, Supp Fig.S4, Supp Fig.S7ab and new table2).

(52)

Fig.1: Indicate the transmembrane part at the maps (possibly class1 only). Consider merging figures 1 and 2.

> We showed it in new Fig2.

Fig. 2c,d: Are the CDTa models in classes 1 and 2 identical?

>

Basically, there are no differences in CDTa between long and short map.

However, as we described before, from 3DVA, we refined two more coordinates for folded CDTa and unfolded CDTa complex. In the folded and unfolded CDTa complex, we built CDTa with whole N-terminal residues as described in method.

(53)

Fig.3: Here, additional panels with surface representation of the NSS loops (hydrophobicity, charge) could reveal more insights into transient binding of CDTa.

> We plotted using hydrophobicity and charge, but it is no more insight unfortunately.

Panel a: show the dimensions of the circles.

> We added the dimensions of the circles in new Fig.4a.

Panel c: "side views" instead of close-up views.

> In new Fig.4c, these are side views of all protomers. We think this is the best way to compare the interactions.

Panel d: Show additionally how the in-conformations of the NSS loop would interact with CTDa. Does this reveal why the conformations are not present in the two subunits?

> We showed these interactions in not only new Fig.4c but also new Supp Fig.S7ab (from folded CDTa and unfolded CDTa complex). We showed the reason why the conformations are biased in C, D and F, which are summarized in the discussion and in new Supp Fig.S7ab.

L. 527(6?): typo.

> Corrected.

(54)

Fig. 4: A comparison of the crystal structure directly with the cryo-Em structure would be more helpful in panel a instead of the per-structure alignment of CDTa and Ia.

> We revised the figures as the reviewer's suggestion. (new Fig. 5)

This is also redundant with SI fig. 1b. c,d: show tilting angle. Define colors for crystal and EM structure in legend.

> Revised.

(55)

Fig. 5: This figure could be merged with Fig. 3, and iota toxin could be shown in the same way like in Fig. 3b. Or, alternatively, show all NSS loop in CDTb, Iota, in prepores and pores, in one SI figure (analogous to SI Fig. 4 and 5).

> We revised this as new Supp Fig.S6 including all NSX-loops of Iota and CDTb. We added pdb IDs in the legend.

The in-conformation is blue, not purple (l. 543).

➤ Corrected.

(56)

Fig. 6: Translocational model could be extended by pH values and assemble of

holotoxin, I would recommend to extend the mechanism according to Fig. 6 in the NSMB 2020 paper of the authors.

> We revised as the new Fig.7, which summarize the scheme from complex formation to translocation.

(57)

Table 1: Give preliminary pdb and EMDB values, and also show the dimer map for completeness.

> We revised the table 1. It includes the dimer map information. It also includes new two maps and coordinates information (the folded and the unfolded CDTa complexes).

(58)

Supplementary data:

SI 1: Typo in legend?;

> Corrected.

Indicate the position of His 314, which is mentioned in the text.

> We added His 314 in SI fig1.

Are the 2ndary structures in b from the crystal structures or from the structure here?

>Those are from cryo-EM structure.

(59)

SI2: Please add map-to-model FSCs, the initial cryosparc model, and more details of the Focused 3D classification.

> We added these information in new Fig.S2.

(60)

SI Fig. 3 could be merged with SI Fig. 2.

> As we reconsidered as all figures, new SI Fig. 3 would be left as the original.

(61)

Si Fig. 5: A side-by-side comparison of all CTDb loops and Iota loops in all investigated conformations would be helpful, possibly merge with SI Fig.

> We revised as new Supp FigS6.

(62)

4. Please also state which ones were processed with C7 symmetry.

We did not use C7 symmetry.

(63)

SI table1typo

> Corrected.

Reviewer #3 (Remarks to the Author):

The results of this study confirm and extend the knowledge about the structure and function of CDTa/CDTb, which was obtained in recent years by other groups, also using cryo-EM as well as further biophysical approaches such as X-ray crystallography and NMR. These former studies revealed a CDTa/CDTb complex with a heptameric structure for CDTb, too, and they also proposed a model for the interaction of CDTb with CDTa and the translocation of CDTa via CDTb. However, in these earlier studies, CDTb was found to be a di-heptamer, while the present work, most likely due to its novel purification protocol for the CDTb pore and a higher resolution, demonstrates that in addition to di-heptamers, CDTb also exists as a single heptamer in the CDTa-bound CDTb pores. This is a central new finding of this manuscript. The authors suggest in their discussion that this is the biologically active CDT complex, which exists in low CDTb concentrations and exhibits toxic effects on cells by delivering

CDTa, while CDTb di-heptamers might exist under high CDTb concentrations to prevent aggregation and/or degradation of this protein.

Based on cryo-EM, the authors performed a detailed analysis of CDTb heptamers and their interaction with CDTa. They found four crucial elements of the CDTb heptamers, i.e. calcium edges that serve for binding of CDTa, NSS-loops that are also involved in CDTa binding and likely crucial for CDTa unfolding and insertion into CDTb pores, a Phe-clamp, which was earlier found in PA63 and Ib heptamers from the related binary anthrax and iota toxins, too, and a stem which forms the channel below the Phe-clamp.

Here, it would be very interesting to investigate the CDTb/CDTa interaction under

acidic conditions, as in endosomes where CDTa translocation through CDTb pores occurs in living cells. This could be included in the manuscript in direct comparison to the interaction under neutral pH conditions.

> Thank you for the comments. We added new SPR experiments in the manuscript and presented the CDTa/CDTb interactions under acidic conditions as well as neutral condition. Furthermore, we also the CDTa/CDTb interactions WT (NSS-loop) and two NSS-loop mutants. Though we did not expect, under acidic conditions, the interaction of CDTa with CDTb-pore did not decrease. We consider it necessary to maintain the interaction before the translocation, even at the acidic pH.

Furthermore, we added two new coordinates folded CDTa and unfolded CDTa from new 3D variability analysis results. It provides one supplementary video which supports the initial unfolding of CDTa in CDTb-pore. We believe these are exciting results.

- 1 Yamada, T. & Tsuge, H. Preparation of *Clostridium perfringens* binary iota-toxin pore complex for structural analysis using cryo-EM. *Methods Enzymol* **649**, 125-148, doi:10.1016/bs.mie.2021.01.032 (2021).
- 2 Anderson, D. M., Sheedlo, M. J., Jensen, J. L. & Lacy, D. B. Structural insights into the transition of *Clostridioides difficile* binary toxin from prepore to pore. *Nat Microbiol* **5**, 102-107, doi:10.1038/s41564-019-0601-8 (2020).
- 3 Sheedlo, M. J., Anderson, D. M., Thomas, A. K. & Lacy, D. B. Structural elucidation of the *Clostridioides difficile* transferase toxin reveals a single-site binding mode for the enzyme. *Proc Natl Acad Sci U S A* **117**, 6139-6144, doi:10.1073/pnas.1920555117 (2020).
- 4 Perelle, S., Gibert, M., Boquet, P. & Popoff, M. R. Characterization of *Clostridium perfringens* Iota-toxin genes and expression in *Escherichia coli*. *Infect Immun* **63**, 4967 (1995).
- 5 Machen, A. J., Fisher, M. T. & Freudenthal, B. D. Anthrax toxin translocation complex reveals insight into the lethal factor unfolding and refolding mechanism. *Sci Rep* **11**, 13038, doi:10.1038/s41598-021-91596-3 (2021).
- 6 Yamada, T. *et al.* Cryo-EM structures reveal translocational unfolding in the clostridial binary iota toxin complex. *Nat Struct Mol Biol* **27**, 288-296, doi:10.1038/s41594-020-0388-6 (2020).

REVIEWERS' COMMENTS

Reviewer #2 (Remarks to the Author):

The authors have addressed my comments and suggestions in an appropriate way. Especially the rearrangement of the figures makes the manuscript more accessible. The additional data, in particular the 3D variability analysis that resulted in a clear description of the initiation of the translocation mechanism by the N-terminal CDTa helix, clearly improve the manuscript and support the author's hypothesis. Therefore, I recommend publication of the revised version.

Minor issues to address:

SI Fig. 4: It would be helpful to label the SPR curves for the individual events (association/dissociation), and also plot the fits into the data curves. Also, reference to Table 2 (and from table to to SI Fig. 4) should be added.

SI Fig. 2: A graphical outline of the 3D variability analysis would also be helpful, probably as an additional SI figure.

Fig. 7: Since the contents of this work do not address the translocation (d), it should be noted and cited.

Reviewer #2 (Remarks to the Author):

The authors have addressed my comments and suggestions in an appropriate way. Especially the rearrangement of the figures makes the manuscript more accessible. The additional data, in particular the 3D variability analysis that resulted in a clear description of the initiation of the translocation mechanism by the N-terminal CDTa helix, clearly improve the manuscript and support the author's hypothesis. Therefore, I recommend publication of the revised version.

Minor issues to address:

(1) SI Fig. 4: It would be helpful to label the SPR curves for the individual events (association/dissociation), and also plot the fits into the data curves. Also, reference to Table 2 (and from table to to SI Fig. 4) should be added.

> We revised SI Fig.4 as the reviewer suggested, adding the plot the fits and triangles indicate the start of association and dissociation, respectively.

We also added the reference in legend in Table2.

SI Fig. 2: A graphical outline of the 3D variability analysis would also be helpful, probably as an additional SI figure.

>

We considered this and confirmed that all information about 3DVA are included in Fig.6, supp Fig.2, and method. Furthermore, supp movie 1 is helpful to understand the 3DVA. So, we leave it without an additional SI figure.

Fig. 7: Since the contents of this work do not address the translocation (d), it should be noted and cited.

>

We added the legend in Fig. 7d as follows. " Unfolded CDTa translocates via the ϕ -clamp, using a Δ pH-driven Brownian ratchet mechanism, though the model is still an open question."